# The Choice of Normalization Influences Shrinkage in Regularized Regression

## Abstract

Regularized models are often sensitive to the scales of the features in the data and it has therefore become standard practice to normalize (center and scale) the features before fitting the model. But there are many different ways to normalize the features and the choice may have dramatic effects on the resulting model. In spite of this, there has so far been no research on this topic. In this paper, we begin to bridge this knowledge gap by studying normalization in the context of lasso, ridge, and elastic net regression. We focus on normal and binary features and show that the class balances of binary features directly influences the regression coefficients and that this effect depends on the combination of normalization and regularization methods used. We demonstrate that this effect can be mitigated by scaling binary features with their variance in the case of the lasso and standard deviation in the case of ridge regression, but that this comes at the cost of increased variance. For the elastic net, we show that scaling the penalty weights, rather than the features, can achieve the same effect. Finally, we also tackle mixes of binary and normal features as well as interactions and provide some initial results on how to normalize features in these cases.

## 1. Introduction

When modeling data where the number of features ($p$) exceeds the number of observations ($n$), it is impossible to apply classical statistical models such as standard linear regression since the design matrix $\boldsymbol{X}$ is no longer of full rank. A common remedy to this problem is to *regularize* the model by adding a penalty term to the objective that punishes models with large coefficients. The resulting problem takes the following form:

$$\underset{\beta_0 \in \mathbb{R}, \boldsymbol{\beta} \in \mathbb{R}^p}{\text{minimize}} \, g(\beta_0, \boldsymbol{\beta}; \boldsymbol{X}, \boldsymbol{y}) + h(\boldsymbol{\beta}), \tag{1}$$

where $g$ is the data-fitting term that attempts to optimize the fit to the data and $h$ is the penalty term that depends only on $\boldsymbol{\beta}$. Two of the most common penalties are the $\ell_1$ norm and squared $\ell_2$ norm penalties, which if $g$ is the standard ordinary least-squares objective, represent the lasso (Tibshirani, 1996; Santosa & Symes, 1986; Donoho & Johnstone, 1994) and ridge (Tikhonov) regression respectively.

These penalties depend on the magnitudes of the coefficients, which means that they are sensitive to the scales of the features in $\boldsymbol{X}$. To avoid this, it is common to *normalize* the features before fitting the model by shifting and scaling each feature by measures of location and scale respectively. For some problems such measures arise naturally from contextual knowledge, but in most cases they must be estimated from data. A popular strategy is to use the mean and standard deviation of each feature as location and scale factors respectively, which is called *standardization*.

The choice of normalization may, however, have consequences for the estimated model. As a first example of this, consider Figure 1, which displays the lasso paths for two data sets and two types of normalization, which yield different results in terms of both estimation and selection of the features.

In spite of this relationship between normalization and regularization, there has so far been no research on the topic. Instead, the choice of normalization is often motivated by computational aspects or by being "standard". Because, although standardization is a natural choice when the features are normally distributed, the choice is not as straightforward for other types of data. In particular, there is no obvious strategy for normalizing binary features (where the each observations takes either of two values) and no previous research that have studied this problem before.

In this paper we begin to bridge this knowledge gap by studying normalization in the context of three particular cases of Equation (1): the lasso, ridge, and elastic net (Zou & Hastie, 2005). The latter of these, the elastic net, is a generalization of the previous two, and is represented by the

---

[1]Anonymous Institution, Anonymous City, Anonymous Region, Anonymous Country. Correspondence to: Anonymous Author <anon.email@domain.com>.

Preliminary work. Under review by the International Conference on Machine Learning (ICML). Do not distribute.

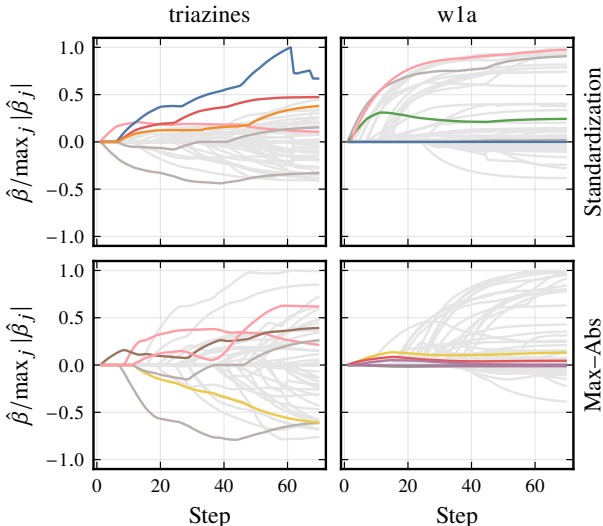

Figure 1. Lasso paths for data sets `triazines` (King, 2024) and `w1a` (Platt, 1998) using standardization and maximum absolute value normalization. (See Appendix E for details on these data sets and Figure 13 in Appendix D.1 for an extended example.) For each data set we color the coefficients if they are among the first five to become non-zero under either normalization scheme.

following optimization problem:

$$\underset{\beta_0 \in \mathbb{R}, \boldsymbol{\beta} \in \mathbb{R}^p}{\text{minimize}} \frac{1}{2} \|\boldsymbol{y} - \beta_0 - \tilde{\boldsymbol{X}}\boldsymbol{\beta}\|_2^2 + \lambda_1 \|\boldsymbol{\beta}\|_1 + \frac{\lambda_2}{2} \|\boldsymbol{\beta}\|_2^2, \quad (2)$$

where setting $\lambda_1 = 0$ results in ridge regression and setting $\lambda_2 = 0$ results in the lasso.

Our focus in this paper is on binary data and we will pay particular attention to the case when binary features are imbalanced, that is, have relatively many ones or zeroes. In this scenario, we will demonstrate that the choice of normalization directly influences the regression coefficients and that this effect depends on the particular choice of normalization and regularization.

## 2. Normalization

There is considerable ambiguity regarding key terms in the literature. Here, we define *normalization* as the process of centering and scaling the feature matrix, which we now formalize.

**Definition 2.1** (Normalization). Let $\boldsymbol{X} \in \mathbb{R}^{n \times p}$ be the feature matrix and let $\boldsymbol{c} \in \mathbb{R}^p$ and $\boldsymbol{s} \in \mathbb{R}_+^p$ be centering and scaling factors respectively. Then $\tilde{\boldsymbol{X}}$ is the *normalized* feature matrix with elements given by $\tilde{x}_{ij} = (x_{ij} - c_j)/s_j$.

There are many different normalization strategies and we have listed a few common choices in Table 1. Standardization is perhaps the most popular type of normalization, at least in the field of statistics. One of its benefits is that it

simplifies certain aspects of fitting the model, such as fitting the intercept. The downside of standardization is that it involves centering by the mean, which destroys sparsity.

Table 1. Common ways to normalize a matrix of features using centering and scaling factors $c_j$ and $s_j$, respectively. Note that $\bar{x}_j$ is the arithmetic mean of feature $j$.

| Normalization | $c_j$ | $s_j$ |
|---|---|---|
| Standardization | $\bar{x}_j$ | $\sqrt{\frac{1}{n} \sum_{i=1}^n (x_{ij} - \bar{x}_j)^2}$ |
| Max–Abs | 0 | $\max_i \|x_{ij}\|$ |
| Min–Max | $\min_i(x_{ij})$ | $\max_i(x_{ij}) - \min_i(x_{ij})$ |
| $\ell_1$-Normalization | 0 or $\bar{x}_j$ | $\|\boldsymbol{x}_j\|_1$ |

When $\boldsymbol{X}$ is sparse, a common alternative to standardization is min–max or max–abs (maximum absolute value) normalization, which scale the data to lie in $[0, 1]$ and $[-1, 1]$ respectively, and retain sparsity when features are binary. These methods are, however, both sensitive to outliers. And since sample extreme values often depend on sample size, as in the case of normal data (Appendix A.1), use of these methods may sometimes be problematic.

## 3. Ridge, Lasso, and Elastic Net Regression

Throughout the paper we assume that the response $\boldsymbol{y}$ is generated according to $y_i = \beta_0^* + \boldsymbol{x}_i^\intercal \boldsymbol{\beta}^* + \varepsilon_i$ for $i \in [n]$ where $[n] = \{1, 2, \ldots, n\}$, with $\boldsymbol{X}$ being the $n \times p$ design matrix with features $\boldsymbol{x}_j$ and where we assume $\boldsymbol{X}$, $\beta_0^*$, and $\boldsymbol{\beta}^*$ to be fixed. We will also assume that the features of the normalized design matrix are orthogonal, that is, $\tilde{\boldsymbol{X}}^\intercal \tilde{\boldsymbol{X}} = \text{diag}\left(\tilde{\boldsymbol{x}}_1^\intercal \tilde{\boldsymbol{x}}_1, \ldots, \tilde{\boldsymbol{x}}_p^\intercal \tilde{\boldsymbol{x}}_p\right)$. In this case, the solution to the coefficients in the elastic net problem is given by

$$\hat{\beta}_j^{(n)} = \frac{S_{\lambda_1}\left(\tilde{\boldsymbol{x}}_j^\intercal \boldsymbol{y}\right)}{\tilde{\boldsymbol{x}}_j^\intercal \tilde{\boldsymbol{x}}_j + \lambda_2}, \qquad \hat{\beta}_0^{(n)} = \frac{\boldsymbol{y}^\intercal \mathbf{1}}{n}, \qquad (3)$$

where $S_\lambda(z)$ is the soft-thresholding operator, defined as $S_\lambda(z) = \text{sign}(z) \max(|z| - \lambda, 0)$. (See Appendix A.2 for a derivation of this.)

Normalization changes the optimization problem and its solution, the coefficients, which will now be on the scale of the normalized features. But we are interested in $\hat{\boldsymbol{\beta}}$: the coefficients on the scale of the original problem. To obtain these, we transform the coefficients from the normalized problem, $\hat{\beta}_j^{(n)}$, back via $\hat{\beta}_j = \hat{\beta}_j^{(n)}/s_j$ for $j \in [p]$. There is a similar transformation for the intercept but we omit here since we are not interested in interpreting it.

Now assume that $\varepsilon_i$ is identically and independently distributed noise with mean zero and finite variance $\sigma_\varepsilon^2$. This

means that the solution is given directly by Equation (3):

$$\hat{\beta}_j = \frac{S_{\lambda_1}(\tilde{\boldsymbol{x}}_j^\mathsf{T}\boldsymbol{y})}{d_j} \quad \text{with} \quad d_j = s_j(\tilde{\boldsymbol{x}}_j^\mathsf{T}\tilde{\boldsymbol{x}}_j + \lambda_2).$$

We can then show that

$$\tilde{\boldsymbol{x}}_j^\mathsf{T}\boldsymbol{y} = \frac{\beta_j^* n\nu_j - \boldsymbol{x}_j^\mathsf{T}\boldsymbol{\varepsilon}}{s_j} \quad \text{and} \quad d_j = s_j\left(\frac{n\nu_j}{s_j^2} + \lambda_2\right), \tag{4}$$

where $\nu_j$ is the uncorrected sample variance of $\boldsymbol{x}_j$. The bias and variance of $\hat{\beta}_j$ are then given by

$$\mathrm{E}\,\hat{\beta}_j - \beta_j^* = \frac{1}{d_j}\,\mathrm{E}\,S_\lambda(\tilde{\boldsymbol{x}}_j^\mathsf{T}\boldsymbol{y}) - \beta_j^*, \tag{5}$$

$$\mathrm{Var}\,\hat{\beta}_j = \frac{1}{d_j^2}\,\mathrm{Var}\,S_\lambda(\tilde{\boldsymbol{x}}_j^\mathsf{T}\boldsymbol{y}). \tag{6}$$

See Appendix A.3 for a derivation of the results above as well as expressions for $\mathrm{E}\,S_\lambda(x)$ and $\mathrm{Var}\,S_\lambda(x)$.

These expressions hold for a general distribution on the error term, provided that its elements are independent and identically distributed. From now on, however, we will add the assumption that $\boldsymbol{\varepsilon}$ is normally distributed, under which both the bias and variance of $\hat{\beta}_j$ have analytical expressions (Appendix A.3.1) and

$$\tilde{\boldsymbol{x}}_j^\mathsf{T}\boldsymbol{y} \sim \mathrm{Normal}\left(\mu_j = \tilde{\boldsymbol{x}}_j^\mathsf{T}\boldsymbol{x}_j\beta_j^*, \sigma_j^2 = \tilde{\boldsymbol{x}}_j^\mathsf{T}\tilde{\boldsymbol{x}}_j\sigma_\varepsilon^2\right).$$

### 3.1. Binary Features

When $x_{ij} \in \{0, 1\}$ for all $i$, we define $\boldsymbol{x}_j$ to be a *binary feature*, and we define the *class balance* of this feature as $q_j = \bar{\boldsymbol{x}}_j$: the proportion of ones. For most of our results, it would make no difference if we were to swap the ones and zeros as long as an intercept is included, and "class balance" is then equivalent to the proportion of either. But in the case of interactions (Section 3.3), the choice matters.

If feature $j$ is binary then $\nu_j = (q_j - q_j^2)$ (the uncorrected sample variance for a binary feature) which in Equation (4) yields

$$\tilde{\boldsymbol{x}}_j^\mathsf{T}\boldsymbol{y} = \frac{\beta_j^* n(q_j - q_j^2) - \boldsymbol{x}_j^\mathsf{T}\boldsymbol{\varepsilon}}{s_j},$$

$$d_j = s_j\left(\frac{n(q_j - q_j^2)}{s_j^2} + \lambda_2\right),$$

and consequently

$$\mu_j = \frac{\beta_j^* n(q_j - q_j^2)}{s_j} \quad \text{and} \quad \sigma_j^2 = \frac{\sigma_\varepsilon^2 n(q_j - q_j^2)}{s_j^2}.$$

We obtain bias and variance of the estimator with respect to $q_j$ by inserting $\tilde{\boldsymbol{x}}_j^\mathsf{T}\boldsymbol{y}$ and $d_j$ into Equations (5) and (6).

The presence of the factor $q_j - q_j^2$ in $\mu_j$, $\sigma_j^2$, and $d_j$ indicates a link between class balance and the elastic net estimator and that this relationship is mediated by the scaling factor $s_j$. To achieve some initial intuition for this relationship, consider the noiseless case ($\sigma_\varepsilon = 0$) in which we have

$$\hat{\beta}_j = \frac{S_{\lambda_1}(\tilde{\boldsymbol{x}}_j^\mathsf{T}\boldsymbol{y})}{s_j\left(\tilde{\boldsymbol{x}}_j^\mathsf{T}\tilde{\boldsymbol{x}}_j + \lambda_2\right)} = \frac{S_{\lambda_1}\left(\frac{\beta_j^* n(q_j - q_j^2)}{s_j}\right)}{s_j\left(\frac{n(q_j - q_j^2)}{s_j^2} + \lambda_2\right)}. \tag{7}$$

This expression shows that class balance directly affects the estimator. For values of $q_j$ close to 0 or 1, the input into the soft-thresholding part of the estimator diminishes and consequently forces the estimate to zero, that is, unless we use the scaling factor $s_j = q_j - q_j^2$, in which case the soft-thresholding part will be unaffected by class imbalance. This choice will not, however, mitigate the impact of class imbalance on the ridge part of the estimator, for which we would instead need $s_j = (q_j - q_j^2)^{1/2}$. For any other choices, $q_j$ will affect the estimator through both the ridge and lasso parts, which means that there exists no scaling $s_j$ that will mitigate this class balance bias in this case. In Section 3.4 we will show how to tackle this issue for the elastic net by scaling the penalty weights. But for now we continue to study the case of normalization.

Based on the reasoning above, we will consider the scaling parameterization $s_j = (q_j - q_j^2)^\delta$, $\delta \geq 0$, which includes the cases that we are primarily interested in, namely $\delta = 0$ (no scaling, as in min–max and max–abs normalization), $\delta = 1/2$ (standard-deviation scaling), and $\delta = 1$ (variance scaling). The last of these, variance scaling, is not a standard type of normalization.

Another consequence of Equation (7), which holds also in the noisy situation, is that normalization affects the estimator even when the binary feature is balanced ($q_j = 1/2$). Using $\delta = 0$, for instance, scales $\beta_j^*$ in the input to $S_\lambda$ by $n(q_j - q_j^2) = n/4$. $\delta = 1$, in contrast, imposes no such scaling in the class-balanced case. And for $\delta = 1/2$, the scaling factor is $n/2$. Generalizing this, we see that to achieve equivalent scaling in the class-balanced case for all types of normalization, under our parameterization, we would need to use $s_j = 4^{\delta-1}(q_j - q_j^2)^\delta$. But this only resolves the issue for the lasso. To achieve a similar effect for ridge regression, we would need another (but similar) modification. When all features are binary, we can just scale $\lambda_1$ and $\lambda_2$ to account for this effect,[1] which is equivalent to modifying $s_j$. But when we consider mixes of binary and normal features in Section 3.2, we need to exert extra care.

We now proceed to consider how class balance affects the bias, variance, and selection probability of the elastic net estimator under the presence of noise. A consequence of

---

[1]We use this strategy in all of the following examples.

our assumption of a normal error distribution and consequent normal distribution of $\tilde{x}_j^\intercal y$ is that the probability of selection in the elastic net problem is given by

$$\Pr\left(\hat{\beta}_j \neq 0\right) =$$
$$\Phi\left(\frac{\beta_j^* n(q_j - q_j^2)^{1/2} - \lambda_1(q_j - q_j^2)^{\delta-1/2}}{\sigma_\varepsilon \sqrt{n}}\right)$$
$$+ \Phi\left(\frac{-\beta_j^* n(q_j - q_j^2)^{1/2} - \lambda_1(q_j - q_j^2)^{\delta-1/2}}{\sigma_\varepsilon \sqrt{n}}\right). \quad (8)$$

Letting $\theta_j = -\mu_j - \lambda_1$ and $\gamma_j = \mu_j - \lambda_1$, we can express this probability asymptotically as $q_j \to 1^+$ as

$$\lim_{q_j \to 1^+} \Pr(\hat{\beta}_j \neq 0) = \begin{cases} 0 & \text{if } 0 \leq \delta < \frac{1}{2}, \\ 2\,\Phi\left(-\frac{\lambda_1}{\sigma_\varepsilon \sqrt{n}}\right) & \text{if } \delta = \frac{1}{2}, \\ 1 & \text{if } \delta > \frac{1}{2}. \end{cases} \quad (9)$$

In Figure 2, we plot this probability for various settings of $\delta$ for a single feature. Our intuition from the noiseless case holds: suitable choices of $\delta$ can mitigate the influence of class imbalance on selection probability. The lower the value of $\delta$, the larger the effect of class imbalance becomes. Note that the probability of selection initially decreases also in the case when $\delta \geq 1$. This is a consequence of increased variance of $\tilde{x}_j^\intercal y$ due to the scaling factor that inflates the noise term. But as $q_j$ approaches 1, the probability eventually rises towards 1 for $\delta \in \{1, 1.5\}$. The reason for this is that this rise in variance eventually quells the soft-thresholding effect altogether. Note, also, that the selection probability is unaffected by $\lambda_2$.

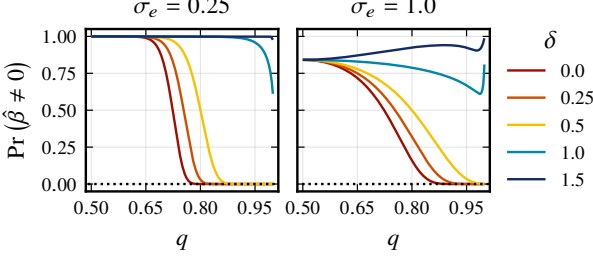

Figure 2. Probability of selection in the lasso given measurement noise $\sigma_\varepsilon$, regularization level $\lambda_1$, and class balance $q$. The scaling factor is set to $s_j = (q - q^2)^\delta$, $\delta \geq 0$. The dotted line represents the asymptotic limit for $\delta = 1/2$ from Equation (9).

Now we turn to the impact of class balance on bias and variance of the elastic net estimator.

**Theorem 3.1.** *If $x_j$ is a binary feature with class balance $q_j \in (0, 1)$, $\lambda_1 \in [0, \infty)$, $\lambda_2 \in [0, \infty)$, $\sigma_\varepsilon > 0$, and $s_j =$*

$(q_j - q_j^2)^\delta$, $\delta \geq 0$ *then*

$$\lim_{q_j \to 1^+} \mathrm{E}\,\hat{\beta}_j = \begin{cases} 0 & \text{if } 0 \leq \delta < \frac{1}{2}, \\ \frac{2n\beta_j^*}{n+\lambda_2}\,\Phi\left(-\frac{\lambda_1}{\sigma_\varepsilon \sqrt{n}}\right) & \text{if } \delta = \frac{1}{2}, \\ \beta_j^* & \text{if } \delta > \frac{1}{2}. \end{cases}$$

**Theorem 3.2.** *Assume the conditions of Theorem 3.1 hold, except that $\lambda_1 > 0$. Then*

$$\lim_{q_j \to 1^+} \mathrm{Var}\,\hat{\beta}_j = \begin{cases} 0 & \text{if } 0 \leq \delta < \frac{1}{2}, \\ \infty & \text{if } \delta \geq \frac{1}{2}. \end{cases}$$

The main take-away of Theorems 3.1 and 3.2 is that there is a bias-variance trade-off with respect to the choice of normalization. We can reduce class imbalance-induced bias by increasing $\delta$ towards 1 (variance scaling) but do so at the price of increase variance. Note that Theorem 3.2 applies only to the case when $\lambda_1 > 1$. In Corollary A.2 (Appendix A.4), we state the corresponding result for ridge regression.

In Figure 3, we now visualize bias, variance, and mean-squared error for ranges of class balance and various noise-level settings for a lasso problem. The figure demonstrates the bias–variance trade-off that our asymptotic results suggest and indicates that the optimal choice of $\delta$ is related to the noise level in the data. Since this level is typically unknown and can only be reliably estimated in the low-dimensional setting, it suggests there might be value in selecting $\delta$ through hyper-optimization. In Figure 12 (Appendix A.6) we show results for ridge regression as well. As expected, it is then $\delta = 1/2$ that leads to unbiased estimates in this case. Also see Figure 11 (Appendix A.6) for extended results on the lasso.

So far, we have only considered a single binary feature, but in Appendix D.2 we present results on power and false discovery rates for a problem with multiple features.

### 3.2. Mixed Data

A fundamental problem with mixes of binary and continuous features is deciding how to put these features on the same scale in order to regularize each type of feature fairly. In principle, we need to match a one-unit change in the binary feature with some amount of change in the normal feature. This problem has previously been tackled, albeit from a different angle, by Gelman (2008), who argued that the common default choice of presenting standardized regression coefficients unduly emphasizes coefficients from continuous features.

To setup this situation formally, we will say that the effects of a binary feature $x_1$ and a normal feature $x_2$ are *comparable* if $\beta_1^* = \kappa\sigma\beta_2^*$, where $\kappa > 0$ represents the number

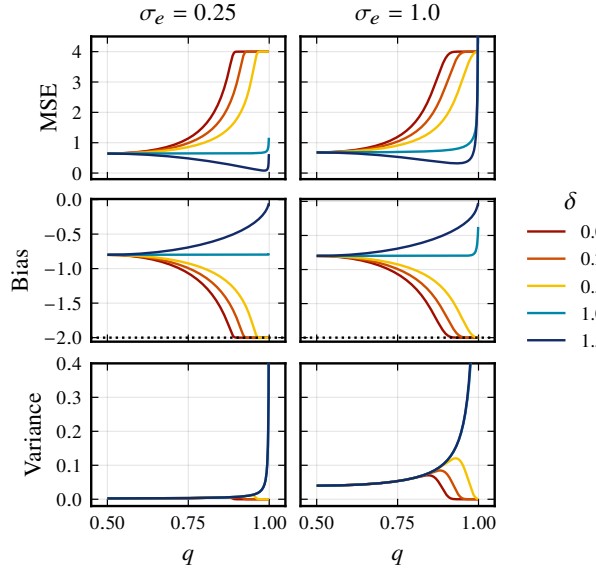

*Figure 3.* Bias, variance, and mean-squared error for a one-dimensional lasso problem, parameterized by noise level ($\sigma_\varepsilon$), class balance ($q$), and scaling ($\delta$). Dotted lines represent asymptotic bias of the lasso estimator in the case when $\delta = 1/2$.

of standard deviations of the normal feature we consider to be comparable to one unit on the binary feature. As an example, assume $\kappa = 2$. Then, if $x_2$ is sampled from Normal $\left(\mu_j, \sigma^2 = (1/2)^2\right)$, the effects of $x_1$ and $x_2$ are comparable if $\beta_1^* = 2\sigma\beta_2^* = \beta_2^*$.

The definition above refers to $\boldsymbol{\beta}^*$, but for our regularized estimates we need $\hat{\beta}_1 = \kappa\sigma\hat{\beta}_2$ to hold. If we assume that we are in a noiseless situation ($\sigma_\varepsilon = 0$), are standardizing the normal feature, and that, without loss of generality, $\bar{x}_1 = 0$, then we need the following equality to hold:

$$\hat{\beta}_1 = \kappa\sigma\hat{\beta}_2 \implies$$

$$\frac{S_{\lambda_1}(\tilde{\boldsymbol{x}}_1^\mathsf{T}\boldsymbol{y})}{s_1\left(\tilde{\boldsymbol{x}}_1^\mathsf{T}\tilde{\boldsymbol{x}}_1 + \lambda_2\right)} = \frac{\kappa\sigma\, S_{\lambda_1}(\tilde{\boldsymbol{x}}_2^\mathsf{T}\boldsymbol{y})}{s_2\left(\tilde{\boldsymbol{x}}_2^\mathsf{T}\tilde{\boldsymbol{x}}_2 + \lambda_2\right)} \implies$$

$$\frac{S_{\lambda_1}\left(\frac{n\beta_1^*(q-q^2)}{s_1}\right)}{s_1\left(\frac{n(q-q^2)}{s_1^2} + \lambda_2\right)} = \frac{\kappa\, S_{\lambda_1}\left(\frac{n\beta_1^*}{\kappa}\right)}{n + \lambda_2}. \quad (10)$$

For the lasso ($\lambda_2 = 0$) and ridge regression ($\lambda_1 = 0$), we see that the equation holds for $s_1 = \kappa(q - q^2)$ and $s_1 = (q - q^2)^{1/2}$, respectively. In other words, we achieve comparability in the lasso by scaling each binary feature with its variance times $\kappa$. And for ridge regression, we can achieve comparability by scaling with standard deviation, irrespective of $\kappa$. For any other choices of $s_1$, equality holds only at a fixed level of class balance. Let this level be $q_0$. Then, to achieve equality for $\lambda_2 = 0$, we need $s_1 = \kappa(q_0 - q_0^2)^{1-\delta}(q - q^2)^\delta$. Similarly, for $\lambda_1 = 0$, we need $s_1 = (q_0 - q_0^2)^{1-2\delta}(q - q^2)^\delta$. In the sequel, we will

assume that $q_0 = 1/2$, to have effects be equivalent for the class-balanced case.

Note that this means that the choice of normalization has an implicit effect on the relative penalization of binary and normal features, even in the class-balanced case ($q_1 = 1/2$). If we for instance use $\delta = 0$ and fit the lasso, then Equation (10) for a binary feature with $q_1 = 1/2$ becomes $4\,S_{\lambda_1}\left(n\beta_1^*/4\right) = \kappa\,S_{\lambda_1}(n\beta_1^*/\kappa)$, which implies $\kappa = 4$. In other words, the choice of normalization equips our model with a belief about how binary and normal features should be penalized relative to one another.

For the rest of this paper, we will use $\kappa = 2$ and say that the effects are comparable if the effect of a flip in the binary feature equals the effect of a two-standard deviation change in the normal feature. We motivate this by an argument by Gelman (2008), but want to stress that the choice of $\kappa$ should, if possible, be based on contextual knowledge of the data and that our results depend only superficially on this particular setting.

### 3.3. Interactions

The elastic net can be extended to include interactions. There is previous literature on this topic (Bien et al., 2013; Zemlianskaia et al., 2022; Lim & Hastie, 2015), but it has not considered the possible influence of normalization. Here, we will consider simple pairwise interactions with no restriction on the presence of main effects. For our analysis, we let $x_1$ and $x_2$ be two features of the data and $x_3$ their interaction, so that $\beta_3$ represents the interaction effect.

We consider two cases in which we assume that the features are orthogonal and that $x_1$ is binary with class balance $q_1$. In the first case, we let $x_2$ be normal with mean $\mu$ and variance $\sigma^2$, and in the second case $x_2$ be binary with class balance $q_2$. To construct the interaction feature, we center[2] the main features and then multiply element-wise. The elements of the interaction feature are then given by $x_{3,i} = (x_{1,i} - \bar{\boldsymbol{x}}_1)(x_{2,i} - \bar{\boldsymbol{x}}_2)$.

If $x_2$ is normal and both features are centered before computing the interaction term, the variance becomes $\sigma^2(q-q^2)$, which suggests using $s_3 = \sigma(q - q^2)^\delta$ along the lines of our previous reasoning. And if $x_2$ is binary, instead, then similar reasoning suggests using $s_3 = ((q_1 - q_1^2)(q_2 - q_2^2))^\delta$. In Section 4.1.4, we study the effects of these choices in simulated experiments.

### 3.4. The Weighted Elastic Net

We have so far shown that certain choices of normalization can mitigate the class-balance bias imposed by the lasso and

---

[2]See Appendix A.5 for motivation for why we center the features before computing the interaction.

ridge regularization. But we have also demonstrated (Section 3.1) that there is no (simple) choice of scaling that can achieve the same effect for the elastic net. Equation (7), however, suggests a natural alternative to normalization, which is to use the weighted elastic net, in which we minimize

$$\frac{1}{2}\|\boldsymbol{y} - \beta_0 - \boldsymbol{X}\boldsymbol{\beta}\|_2^2 + \lambda_1 \sum_{j=1}^p u_j |\beta_j| + \frac{\lambda_2}{2} \sum_{j=1}^p v_j \beta_j^2,$$

with $\boldsymbol{u}$ and $\boldsymbol{v}$ being $p$-length vectors of positive scaling factors. This is equivalent to the standard elastic net for a normalized feature matrix when $u_j = s_j$ and $v_j = s_j^2$, which can be seen by substituting $\beta_j s_j = \tilde{\beta}_j$ in Equation (2) and solving for $\tilde{\boldsymbol{\beta}}$. Note that we do not need to rescale the coefficients from this problem as we would for the standard elastic net on normalized data.

This allows us to control class-balance bias by setting our weights according to $u_j = v_j = (q_j - q_j^2)^\omega$ and counteract it, at least in the noiseless case, with $\omega = 1$, which, we want to emphasize, is *not* possible using the standard elastic net. For the lasso and ridge regression, however, this setting of $\omega = 1$ is equivalent to using $\delta = 1$ and $\delta = 1/2$, respectively, in the standard elastic net with normalized data.

Results analogous to those in Section 3.1 can be attained with a few small modifications for the weighted elastic net case. Starting with selection probability, we can set $s_j = 1$ and replace $\lambda_1$ with $\lambda_1 u_j = \lambda_1 (q_j - q_j^2)^\omega$ in Equation (8), which shows that $\omega$ and $\delta$ have interchangeable effects for selection probability.

As far as expected value and variance of the weighted elastic net estimator is concerned, the same expressions apply directly in the case of the weighted elastic net given $s_j = 1$ for all $j$ and replacing $\lambda_1$ as in the previous paragraph and $\lambda_2$ with $\lambda_2(q_j - q_j^2)^\omega$. On the other hand, the asymptotic results differ slightly as we now show.

**Theorem 3.3.** *Let $\boldsymbol{x}_j$ be a binary feature with class balance $q_j \in (0, 1)$ and take $\lambda_1 > 0$, $\lambda_2 > 0$, and $\sigma_\varepsilon > 0$. For the weighted elastic net with weights $u_j = v_j = (q_j - q_j^2)^\omega$ and $\omega \geq 0$, it holds that*

$$\lim_{q_j \to 1^+} \mathrm{E}\, \hat{\beta}_j = \begin{cases} 0 & \text{if } 0 \leq \omega < 1, \\ \frac{\beta^* n}{n + \lambda_2} & \text{if } \omega = 1, \\ \beta^* & \text{if } \omega > 1, \end{cases}$$

$$\lim_{q_j \to 1^+} \mathrm{Var}\, \hat{\beta}_j = \begin{cases} 0 & \text{if } 0 \leq \delta < \frac{1}{2}, \\ \infty & \text{if } \delta \geq \frac{1}{2}. \end{cases}$$

This result for expected value is similar to the one for the unweighted but normalized elastic net. The only difference arises in the case when $\omega = 1$, in which case the limit is unaffected by $\lambda_1$ in the case of the weighted elastic net. For

variance, the result mimics the result for the elastic net with normalization.

The results for bias, variance, and mean-squared error for the weighted elastic net are similar to those in Figure 3 and are plotted in Figure 10 (Appendix A.6).

## 4. Experiments

In the following sections we present the results of our experiments. For all simulated data we generate our response vector according to $\boldsymbol{y} = \boldsymbol{X}\boldsymbol{\beta}^* + \boldsymbol{\varepsilon}$, with $\boldsymbol{\varepsilon} \sim \mathrm{Normal}(\boldsymbol{0}, \sigma_\varepsilon^2 \boldsymbol{I})$. We consider two types of features: binary (quasi-Bernoulli) and quasi-normal features. To generate binary vectors, we sample $\lceil n q_j \rceil$ indexes uniformly at random without replacement from $[n]$ and set the corresponding elements to one and the remaining ones to zero. To generate quasi-normal features, we generate a linear sequence $\boldsymbol{w}$ with $n$ values from $10^{-4}$ to $1 - 10^{-4}$, set $x_{ij} = \Phi^{-1}(w_i)$, and then shuffle the elements of $\boldsymbol{x}_j$ uniformly at random.

We use a coordinate descent solver to optimize our models, which we have based on the algorithm outlined by Friedman et al. (2010). All experiments were coded using the Julia programming language (Bezanson et al., 2017) and the code is available in the supplementary material. All simulated experiments were run for at least 100 iterations and, unless stated otherwise, are presented as means $\pm$ one standard deviation (using bars or ribbons).

### 4.1. Normalization in the Lasso and Ridge Regression

In this section we consider fitting the lasso and ridge regression to normalized data sets. To normalize the data, we use standardize all quasi-normal features. For binary features, we center by mean and scale by $s_j \propto (q_j - q_j^2)^\delta$.

#### 4.1.1. VARIABILITY AND BIAS IN ESTIMATES

In our first experiment, we consider fitting the lasso to a simulated data set with $n = 500$ observations and $p = 1000$ features, out of which the first 20 features correspond to signals, with $\beta_j^*$ decreasing linearly from 1 to 0.1. We introduce dependence between the features by copying the first $\lceil \rho n/2 \rceil$ values from the first feature to each of the following features. In addition, we set the class balance of the first 20 features so that it decreases linearly on a log-scale from 0.5 to 0.99. We estimate the regression coefficients using the lasso, setting $\lambda_1 = 2\sigma_\varepsilon \sqrt{2 \log p}$.

The results (Figure 4, and Figure 15 in Appendix A.6) show that class balance has considerable effect, particularly in the case of no scaling ($\delta = 0$), which corroborates our theory from Section 3.1. At $q_j = 0.99$, for instance, the estimate ($\hat{\beta}_{20}$) is consistently zero when $\delta = 0$. For $\delta = 1$, we see that class imbalance increases the variance of the

estimates. What is also clear is that the variance of the estimates increase with class imbalance and that this effect increases together with $\delta$.

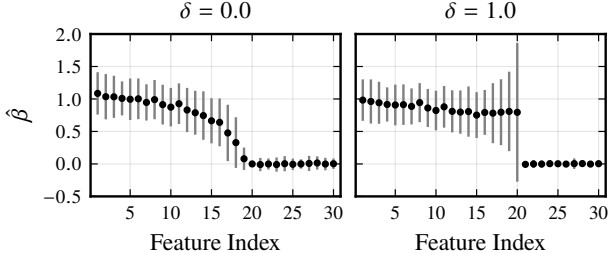

*Figure 4.* Regression coefficients for a lasso problem with binary data where $n = 500$ and $p = 1000$ with 20 true signals. Here we show only the first 30 coefficients. See Section 4.1 for more information on the setup of this experiment.

### 4.1.2. PREDICTIVE PERFORMANCE

In this section we examine the influence of normalization on predictive performance for three different data sets: a1a (Becker & Kohavi, 1996), rhee2006 (Rhee et al., 2006), and w1a (Platt, 1998).[3] We evaluate performance in terms of normalized mean-squared error (NMSE) for lasso and ridge regression across a two-dimensional grid of $\delta$ and $\lambda$, where for $\delta$ we use a linear sequence from 0 to 1, and for $\lambda$ a geometric sequence from $\lambda_{\max}$ (the value of $\lambda$ at which the first feature enters the model) to $10^{-2}\lambda_{\max}$. We split the data into equal training and validation set splits and for each combination of $\lambda$ and $\delta$ fit the lasso or ridge to the training set.

We present the results for ridge regression in Figure 5, which shows contour plots of the validation set error. We see that optimal setting of $\delta$ differs between the different data sets, suggesting that it is useful to choose $\delta$ by hyperparameter optimization. See Figure 16 (Appendix D.2.1) for a plot that includes the lasso as well.

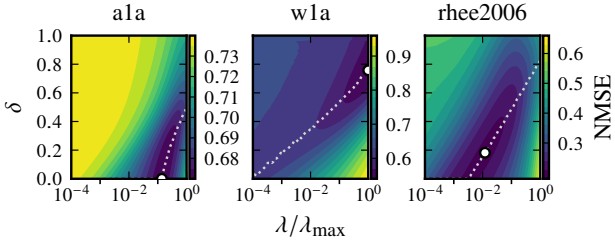

*Figure 5.* Contour plots of normalized validation set mean-squared error (NMSE) for $\delta$ and $\lambda$ in ridge regression on three real data sets. The dotted path shows the smallest NMSE as a function of $\lambda$ and the circles mark combinations with the lowest error.

In Appendix D.2.1, we extend these results with experiments

---

[3]See Appendix E for details about these data sets.

on simulated data under various class balances and signal-to-noise ratios, again showing that normalization has an impact on predictive preformance.

### 4.1.3. MIXED DATA

In Section 3.2 we showed theoretically that care needs to be taken when normalizing mixed data. Here we verify the theory through simulations. We construct a quasi-normal feature with mean zero and standard deviation $1/2$ and a binary feature with varying class balance $q_j$. We set the signal-to-noise ratio to 0.5 and use $n = 1000$. These features are constructed so that their effects are comparable under the notion of comparability that we introduced in Section 3.2 using $\kappa = 2$. In order to preserve the comparability for the baseline case when we have perfect class balance, we scale by $s_j = 2 \times (1/4)^{1-\delta}(q_j - q_j^2)^\delta$. Finally, we set $\lambda$ to $\lambda_{\max}/2$ and $2\lambda_{\max}$ for lasso and ridge regression respectively.

The results (Figure 6) reflect our theoretical results from Section 3. In the case of the lasso, we need $\delta = 1$ (variance scaling) to avoid the effect of class imbalance, whereas for ridge we instead need $\delta = 1/2$ (standardization). As our theory suggests, this extra scaling mitigates this class-balance dependency at the cost of added variance.

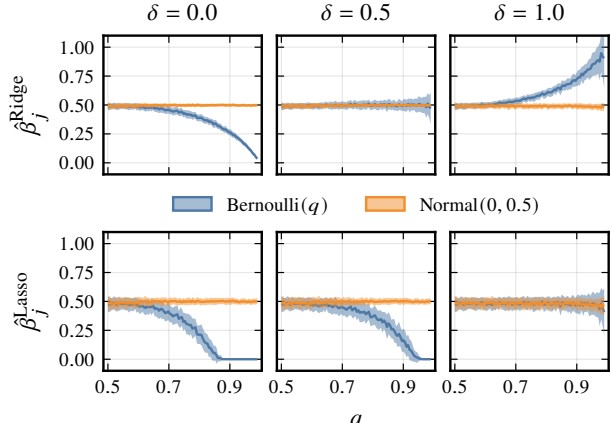

*Figure 6.* Lasso and ridge estimates for a two-dimensional problem where one feature is a binary feature with class balance $q_j$ (Bernoulli($q_j$)) and the other is quasi-normal with standard deviation $1/2$, (Normal(0, 0.5)).

### 4.1.4. INTERACTIONS

Next, we study the effects of normalization and class balance on interactions in the lasso. Our example consists of a two-feature problem with an added interaction term given by $x_{i3} = x_{i1}x_{i2}$. The first feature is binary with class balance $q$ and the second quasi-normal with standard deviation 0.5. We use $n = 1000$, $\lambda_1 = n/4$, and normalize the binary feature by mean-centering and scaling by $\kappa(q - q^2)$, using

$\kappa = 2$. We consider two different strategies for choosing $s_3$: in the first strategy, which we call *Strategy 1*, we simply standardize the resulting interaction feature. In the second strategy, *Strategy 2* we center with mean and scale with $s_1 s_2$ (the product of the scales of the binary and normal features).

The results for $\boldsymbol{\beta}^* = \mathbf{1}$ (Figure 7) show that only strategy 2 estimates the effect of the interaction correctly. Strategy 1, meanwhile, only selects the correct model if the class balance of the binary feature is close to 1/2 and in general shrinks the coefficient too much. See Figure 19 (Appendix D.3) for results on different choices of $\boldsymbol{\beta}^*$.

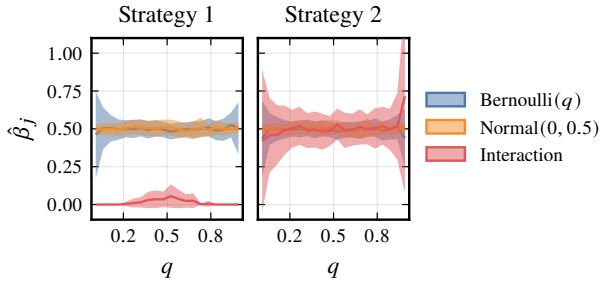

*Figure 7.* Lasso estimates for a problem with a binary feature, a quasi-normal feature, and an interaction feature. We have set $\boldsymbol{\beta}^* = \mathbf{1}$ and use two different normalization strategies where Strategy 1 represents standardization and Strategy 2 is mean-centering together with scaling by $s_1 s_2$.

### 4.2. The Weighted Elastic Net

The weighted elastic net can be used as an alternative to normalization to correct for class balance bias when $\lambda_1 > 0$ and $\lambda_2 > 0$. To simplify the presentation, we parameterize the elastic net as $\lambda_1 = \alpha\lambda$ and $\lambda_2 = (1 - \alpha)\lambda$, so that $\alpha$ controls the balance between the ridge and lasso. We conduct an experiment with the same setup as in Section 4.1.3, but here we use the weighted elastic net instead with $\alpha = 0.5$ (See Appendix D.4 for results using other setting for $\alpha$). We use $n = 1000$ and vary $\omega$, using the weights $u_j = v_j = (q_j - q_j^2)^\omega$ as we suggested in Section 3.4. Our results (Figure 8) show that $\omega = 1$ leads to seemingly unbiased estimates.

## 5. Discussion

We have studied the effects of normalization in lasso, ridge, and elastic net regression with binary data—an issue that has not been studied. We discovered that the class balance (proportion of ones) of these binary features has a pronounced effect on both lasso and ridge estimates and that this effect depends on the type of normalization used. For the lasso, for instance, features with large class imbalances stand little chance of being selected if the features are standardized, even when their relationships with the response are strong.

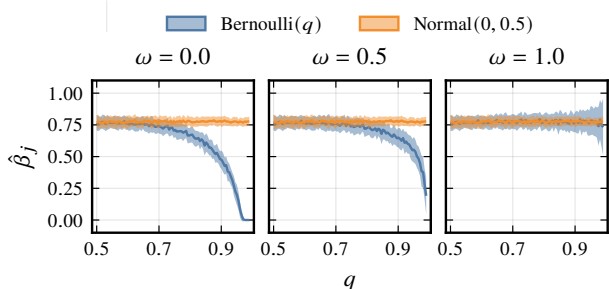

*Figure 8.* Weighted elastic net estimates for $\alpha = 0.5$ for a problem with a binary feature with class balance $q$ (Bernoulli($q$)) and quasi-normal with standard deviation 1/2 (Normal($0, 0.5$)). $\omega$ indicates the scaling of the penalty weights.

The driver of this result is the relationship between the variance of the feature and type of normalization. This works as expected for normally distributed features. But for binary features it means that a one-unit change is treated differently depending on the corresponding feature's class balance, which we believe may surprise some. We have, however, shown that scaling binary features with standard deviation in the case of ridge regression and variance in the case of the lasso mitigates this effect, but that doing so comes at the price of increased variance. This effectively means that the choice of normalization constitutes a bias–variance trade-off.

We have also studied the case of mixed data: designs that include both binary and normally distributed features (Section 3.2). In this setting, our first finding is that there is an implicit relationship between the choice of normalization and the manner in which regularization affects binary viz-a-viz normally distributed features, even when the binary feature is perfectly balanced. The choice of max–abs normalization, for instance, leads to a specific weighting of the effects of binary features relative to those of normal features.

For interactions between binary and normal features features (Section 4.1.4), our conclusions are that the interaction feature should be computed after centering both the binary and normal feature and that scaling with the product of the standard deviation of the normal feature and variance of the binary features mitigates the class balance bias in this case.

Finally, note that our theoretical results are limited by several assumptions: 1) a fixed feature matrix $\boldsymbol{X}$, 2) orthogonal features, and 3) normal and independent errors. In future studies, it would be interesting to relax these assumptions and study the effects of normalization in a more general setting. We have also focused on the case of binary and continuous features here, but are convinced that categorical features are also of interest and might raise additional challenges with respect to normalization.

## Impact Statement

This paper presents work whose goal is to advance the field of Machine Learning. There are many potential societal consequences of our work, none which we feel must be specifically highlighted here.

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

## A. Additional Theory

### A.1. Maximum–Absolute and Min–Max Normalization for Normally Distributed Data

In Theorem A.1, we show that the scaling factor in the max–abs method converges in distribution to a Gumbel distribution.

**Theorem A.1.** *Let $X_1, X_2, \ldots, X_n$ be a sample of normally distributed random variables, each with mean $\mu$ and standard deviation $\sigma$. Then*

$$\lim_{n \to \infty} \Pr\left(\max_{i \in [n]} |X_i| \leq x\right) = G(x),$$

*where $G$ is the cumulative distribution function of a Gumbel distribution with parameters*

$$b_n = F_Y^{-1}(1 - 1/n) \quad \text{and} \quad a_n = \frac{1}{n f_Y(\mu_n)},$$

*where $f_Y$ and $F_Y^{-1}$ are the probability distribution function and quantile function, respectively, of a folded normal distribution with mean $\mu$ and standard deviation $\sigma$.*

The gist of Theorem A.1 is that the limiting distribution of $\max_{i \in [n]} |X_i|$ has expected value $b_n + \gamma a_n$, where $\gamma$ is the Euler-Mascheroni constant, which shows that the scaling factor depends on the sample size. In Figure 9(a), we observe empirically that the limiting distribution agrees well with the empirical distribution in expected value even for small values of $n$.

In Figure 9(b) we show the effect of increasing the number of observations, $n$, in a two-feature lasso model with max-abs normalization applied to both features. The coefficient corresponding to the Normally distributed feature shrinks as the number of observation $n$ increases. Since the expected value of the Gumbel distribution diverges with $n$, this means that there's always a large enough $n$ to force the coefficient in a lasso problem to zero with high probability.

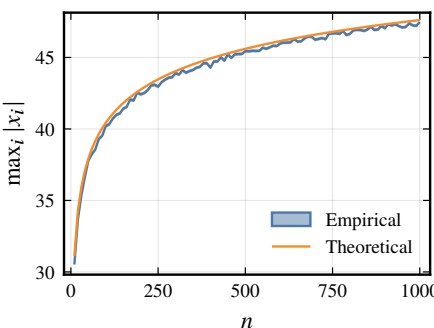
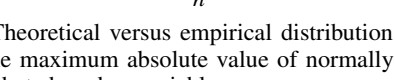
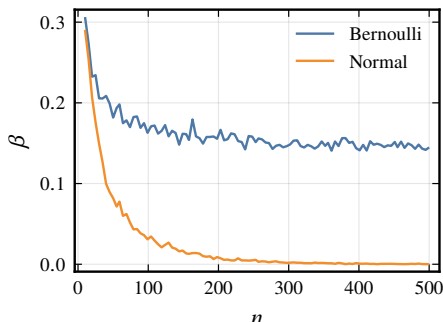

(a) Theoretical versus empirical distribution of the maximum absolute value of normally distributed random variables.

(b) Estimation of mixed features under maximum absolute value scaling

*Figure 9.* Effects of maximum absolute value scaling

For min–max normalization, the situation is similar and we omit the details here. The main point is that the scaling factor is strongly dependent on the sample size, which makes it unsuitable for normally distributed data in several situations, such as on-line learning (where sample size changes over time) or model validation with uneven data splits.

### A.2. Solution to the Elastic Net

Let $(\hat{\beta}_0^{(n)}, \hat{\boldsymbol{\beta}}^{(n)})$ be a solution to the problem in Equation (2). Expanding the function, we have

$$\frac{1}{2}\left(\boldsymbol{y}^\intercal \boldsymbol{y} - 2(\tilde{\boldsymbol{X}}\boldsymbol{\beta} + \beta_0)^\intercal \boldsymbol{y} + (\tilde{\boldsymbol{X}}\boldsymbol{\beta} + \beta_0)^\intercal(\tilde{\boldsymbol{X}}\boldsymbol{\beta} + \beta_0)\right) + \lambda_1\|\boldsymbol{\beta}\|_1 + \frac{\lambda_2}{2}\|\boldsymbol{\beta}\|_2^2.$$

Taking the subdifferential with respect to $\boldsymbol{\beta}$ and $\beta_0$, the KKT stationarity condition yields the following system of equations:

$$\begin{cases} \tilde{\boldsymbol{X}}^\intercal(\tilde{\boldsymbol{X}}\boldsymbol{\beta} + \beta_0 - \boldsymbol{y}) + \lambda_1 g + \lambda_2 \boldsymbol{\beta} \ni \boldsymbol{0}, \\ n\beta_0 + (\tilde{\boldsymbol{X}}\boldsymbol{\beta})^\intercal \boldsymbol{1} - \boldsymbol{y}^\intercal \boldsymbol{1} = 0, \end{cases} \tag{11}$$

where $g$ is a subgradient of the $\ell_1$ norm that has elements $g_i$ such that

$$g_i \in \begin{cases} \{\operatorname{sign}\beta_i\} & \text{if } \beta_i \neq 0, \\ [-1, 1] & \text{otherwise.} \end{cases}$$

### A.2.1. ORTHOGONAL FEATURES

If the features of the normalized design matrix are orthogonal, that is, $\tilde{X}^\mathsf{T}\tilde{X} = \operatorname{diag}\left(\tilde{x}_1^\mathsf{T}\tilde{x}_1, \ldots, \tilde{x}_p^\mathsf{T}\tilde{x}_p\right)$, then Equation (11) can be decomposed into a set of $p + 1$ conditions:

$$\begin{cases} \tilde{x}_j^\mathsf{T}(\tilde{x}_j\beta_j + \mathbf{1}\beta_0 - y) + \lambda_2\beta_j + \lambda_1 g \ni 0, & j \in [p], \\ n\beta_0 + (\tilde{X}\beta)^\mathsf{T}\mathbf{1} - y^\mathsf{T}\mathbf{1} = 0. \end{cases}$$

The inclusion of the intercept ensures that the locations (means) of the features do not affect the solution (except for the intercept itself). We will therefore from now on assume that the features are mean-centered so that $c_j = \bar{x}_j$ for all $j$ and therefore $\tilde{x}_j^\mathsf{T}\mathbf{1} = 0$. A solution to the system of equations is then given by the following set of equations (Donoho & Johnstone, 1994):

$$\hat{\beta}_j^{(n)} = \frac{\mathrm{S}_{\lambda_1}\left(\tilde{x}_j^\mathsf{T}y\right)}{\tilde{x}_j^\mathsf{T}\tilde{x}_j + \lambda_2}, \qquad \hat{\beta}_0^{(n)} = \frac{y^\mathsf{T}\mathbf{1}}{n},$$

where $\mathrm{S}_\lambda(z) = \operatorname{sign}(z)\max(|z| - \lambda, 0)$ is the soft-thresholding operator.

### A.3. Bias and Variance of the Elastic Net Estimator

Here, we derive the results in Section 3 in more detail. Let

$$Z_j = \tilde{x}_j^\mathsf{T}y = \tilde{x}_j^\mathsf{T}(X\beta^* + \varepsilon) = \tilde{x}_j^\mathsf{T}(x_j\beta_j^* + \varepsilon) \qquad \text{and} \qquad d_j = s_j(\tilde{x}_j^\mathsf{T}\tilde{x}_j + \lambda_2)$$

so that $\hat{\beta}_j = \mathrm{S}_{\lambda_1}(Z_j)/d_j$. Since $d_j$ is fixed under our assumptions, we focus on $S_{\lambda_1}(Z_j)$. First observe that since $c_j = \bar{x}_j$,

$$\tilde{x}_j^\mathsf{T}\tilde{x}_j = \frac{1}{s_j^2}(x_j - c_j)^\mathsf{T}(x_j - c_j) = \frac{x_j^\mathsf{T}x_j - nc_j^2}{s_j^2} = \frac{n\nu_j}{s_j^2},$$

$$\tilde{x}_j^\mathsf{T}x_j = \frac{1}{s_j}(x_j^\mathsf{T}x_j - x_j^\mathsf{T}\mathbf{1}c_j) = \frac{n\nu_j}{s_j},$$

where $\nu_j$ is the uncorrected sample variance of $x_j$. This means that

$$Z_j = \frac{\beta_j^* n\nu_j - x_j^\mathsf{T}\varepsilon}{s_j} \qquad \text{and} \qquad d_j = s_j\left(\frac{n\nu_j}{s_j^2} + \lambda_2\right). \tag{12}$$

For the expected value and variance of $Z_j$ we then have

$$\mathrm{E}\,Z_j = \mu_j = \mathrm{E}\left(\tilde{x}_j^\mathsf{T}(x_j\beta_j^* + \varepsilon)\right) = \tilde{x}_j^\mathsf{T}x_j\beta_j^* = \frac{\beta_j^* n\nu_j}{s_j},$$

$$\operatorname{Var}Z_j = \sigma_j^2 = \operatorname{Var}\left(\tilde{x}_j^\mathsf{T}\varepsilon\right) = \tilde{x}_j^\mathsf{T}\tilde{x}_j\sigma_\varepsilon^2 = \frac{n\nu_j\sigma_\varepsilon^2}{s_j^2}.$$

The expected value of the soft-thresholding estimator is

$$\mathrm{E}\,\mathrm{S}_\lambda(Z_j) = \int_{-\infty}^\infty \mathrm{S}_\lambda(z)f_{Z_j}(z)\,\mathrm{d}z = \int_{-\infty}^{-\lambda}(z + \lambda)f_{Z_j}(z)\,\mathrm{d}z + \int_\lambda^\infty (z - \lambda)f_{Z_j}(z)\,\mathrm{d}z.$$

And then the bias of $\hat{\beta}_j$ with respect to the true coefficient $\beta_j^*$ is

$$\mathrm{E}\,\hat{\beta}_j - \beta_j^* = \frac{1}{d_j}\,\mathrm{E}\,\mathrm{S}_\lambda(Z_j) - \beta_j^*.$$

Finally, we note that the variance of the soft-thresholding estimator is

$$\operatorname{Var} S_\lambda(Z_j) = \int_{-\infty}^{-\lambda} (z + \lambda)^2 f_{Z_j}(z)\,\mathrm{d}z + \int_{\lambda}^{\infty} (z - \lambda)^2 f_{Z_j}(z)\,\mathrm{d}z - \left(\operatorname{E} S_\lambda(Z_j)\right)^2 \tag{13}$$

and that the variance of the elastic net estimator is therefore

$$\operatorname{Var} \hat{\beta}_j = \frac{1}{d_j^2} \operatorname{Var} S_\lambda(Z_j).$$

### A.3.1. NORMALLY DISTRIBUTED NOISE

We now assume that $\varepsilon$ is normally distributed. Then

$$Z_j \sim \operatorname{Normal}\left(\mu_j = \tilde{\boldsymbol{x}}_j^\mathsf{T} \boldsymbol{x}_j \beta_j^*, \sigma_j^2 = \tilde{\boldsymbol{x}}_j^\mathsf{T} \tilde{\boldsymbol{x}}_j \sigma_\varepsilon^2\right).$$

Let $\theta_j = -\mu_j - \lambda_1$ and $\gamma_j = \mu_j - \lambda_1$. Then the expected value of soft-thresholding of $Z_j$ is

$$\operatorname{E} S_{\lambda_1}(Z_j) = \int_{-\infty}^{\frac{\theta_j}{\sigma_j}} (\sigma_j u - \theta_j)\,\phi(u)\,\mathrm{d}u + \int_{-\frac{\gamma_j}{\sigma_j}}^{\infty} (\sigma_j u + \gamma_j)\,\phi(u)\,\mathrm{d}u$$

$$= -\theta_j\,\Phi\left(\frac{\theta_j}{\sigma_j}\right) - \sigma_j\,\phi\left(\frac{\theta_j}{\sigma_j}\right) + \gamma_j\,\Phi\left(\frac{\gamma_j}{\sigma_j}\right) + \sigma_j\,\phi\left(\frac{\gamma_j}{\sigma_j}\right)$$

where $\phi(u)$ and $\Phi(u)$ are the probability density and cumulative distribution functions of the standard normal distribution, respectively. Computing Equation (13) gives us

$$\operatorname{Var} S_\lambda(Z_j) = \frac{\sigma_j^2}{2} \left( \operatorname{erf}\left(\frac{\theta_j}{\sigma_j\sqrt{2}}\right) \quad \frac{\theta_j}{\sigma_j}\sqrt{\frac{2}{\pi}} \exp\left(-\frac{\theta_j^2}{2\sigma_j^2}\right) + 1 \right)$$

$$+ 2\theta_j\sigma_j\,\phi\left(\frac{\theta_j}{\sigma_j}\right) + \theta_j^2\,\Phi\left(\frac{\theta_j}{\sigma_j}\right)$$

$$+ \frac{\sigma_j^2}{2} \left( \operatorname{erf}\left(\frac{\gamma_j}{\sigma_j\sqrt{2}}\right) - \frac{\gamma_j}{\sigma_j}\sqrt{\frac{2}{\pi}} \exp\left(-\frac{\gamma_j^2}{2\sigma_j^2}\right) + 1 \right)$$

$$+ 2\gamma_j\sigma_j\,\phi\left(\frac{\gamma_j}{\sigma_j}\right) + \gamma_j^2\,\Phi\left(\frac{\gamma_j}{\sigma_j}\right)$$

$$- \left(\operatorname{E} S_{\lambda_1}(Z_j)\right)^2.$$

### A.4. Bias and Variance for Ridge Regression

**Corollary A.2** (Variance in Ridge Regression). *Assume the conditions of Theorem 3.1 hold, except that $\lambda_1 = 0$. Then*

$$\lim_{q_j \to 1^+} \operatorname{Var} \hat{\beta}_j = \begin{cases} 0 & \text{if } 0 \le \delta < 1/4, \\ \frac{\sigma_\varepsilon^2 n}{\lambda_2^2} & \text{if } \delta = 1/4, \\ \infty & \text{if } \delta > 1/4. \end{cases}$$

### A.5. Centering and Interaction Features

The main motivation for centering is that it removes correlation between the main features and the interaction, which would otherwise affect the estimates due to the regularization. Centering normal features is also important because it ensures that their means do not factor into the estimation of their effects, which is otherwise the case since the variance of $\boldsymbol{x}_3$ would then be $q_1(\sigma^2 + \mu^2(1 - q_1))$ in the case when $\boldsymbol{x}_1$ is centered and $(q_1 - q_1^2)(\sigma^2 + \mu^2)$ otherwise. Centering binary features is also important because the variance of the interaction term is otherwise $q_1\sigma^2$ (provided $\boldsymbol{x}_2$ is centered), which would mean that the encoding of values of the binary feature (e.g. $\{0, 1\}$ versus $\{-1, 1\}$) would affect the interaction term.

### A.6. Extended Results on Bias and Variance for Ridge, Lasso, and Elastic Net Regression

In Figure 10, we show bias, variance, and mean-squared error for the weighted elastic net. We see that the behavior of bias as $q_j \to 1^+$ depends on noise level and that there is a bias–variance trade-off with respect to $\omega$. As in Section 3.2, we modify the weighting factor to have comparability under $\kappa = 2$.

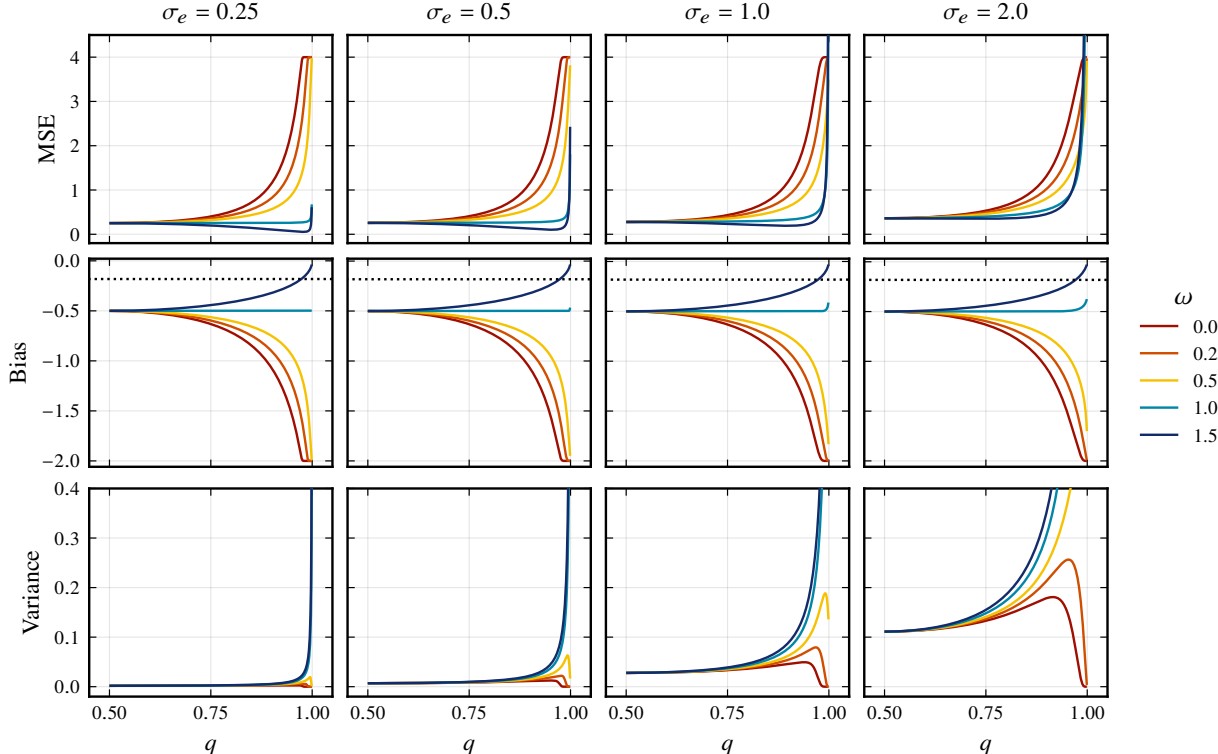

*Figure 10.* Bias, variance, and mean-squared error in the case of the one-dimensional weighted elastic net. The measures are shown for different noise levels ($\sigma_\varepsilon$), class balances ($q_j$), and values of ($\omega$), which controls the weights that are set to $u_j = v_j = 2 \times 4^{\omega-1}(q - q^2)^\omega$ in order for the results to be comparable across different values of $\omega$. The dotted lines represent the asymptotic bias of the estimator in the case of $\omega = 1$. In the case of $\omega > 1$, the limit of the bias is zero.

## B. Proofs

## C. Proof of Theorem 3.1

To avoid excessive notation, we allow ourselves to abuse notation and will drop the subscript $j$ everywhere in this proof, allowing $\beta^*$, $s$, and so on to respectively denote $\beta^*$, $s_j$ et cetera.

Since $s = (q - q^2)^\delta$, we have

$$\mu = \beta^* n(q - q^2)^{1-\delta}, \qquad \frac{\theta}{\sigma} = -a\sqrt{q - q^2} - b(q - q^2)^{\delta-1/2},$$

$$\sigma = \sigma_\varepsilon\sqrt{n}(q - q^2)^{1/2-\delta}, \qquad \frac{\gamma}{\sigma} = a\sqrt{q - q^2} - b(q - q^2)^{\delta-1/2},$$

$$d = n(q - q^2)^{1-\delta} + \lambda_2(q - q^2)^\delta, \qquad \frac{\theta}{d} = \frac{-\beta^* n - \lambda_1(q - q^2)^{\delta-1}}{n + \lambda_2(q - q^2)^{2\delta-1}},$$

$$\theta = -\beta^* n(q - q^2)^{1-\delta} - \lambda_1, \qquad \frac{\gamma}{d} = \frac{\beta^* n - \lambda_1(q - q^2)^{\delta-1}}{n + \lambda_2(q - q^2)^{2\delta-1}},$$

$$\gamma = \beta^* n(q - q^2)^{1-\delta} - \lambda_1,$$

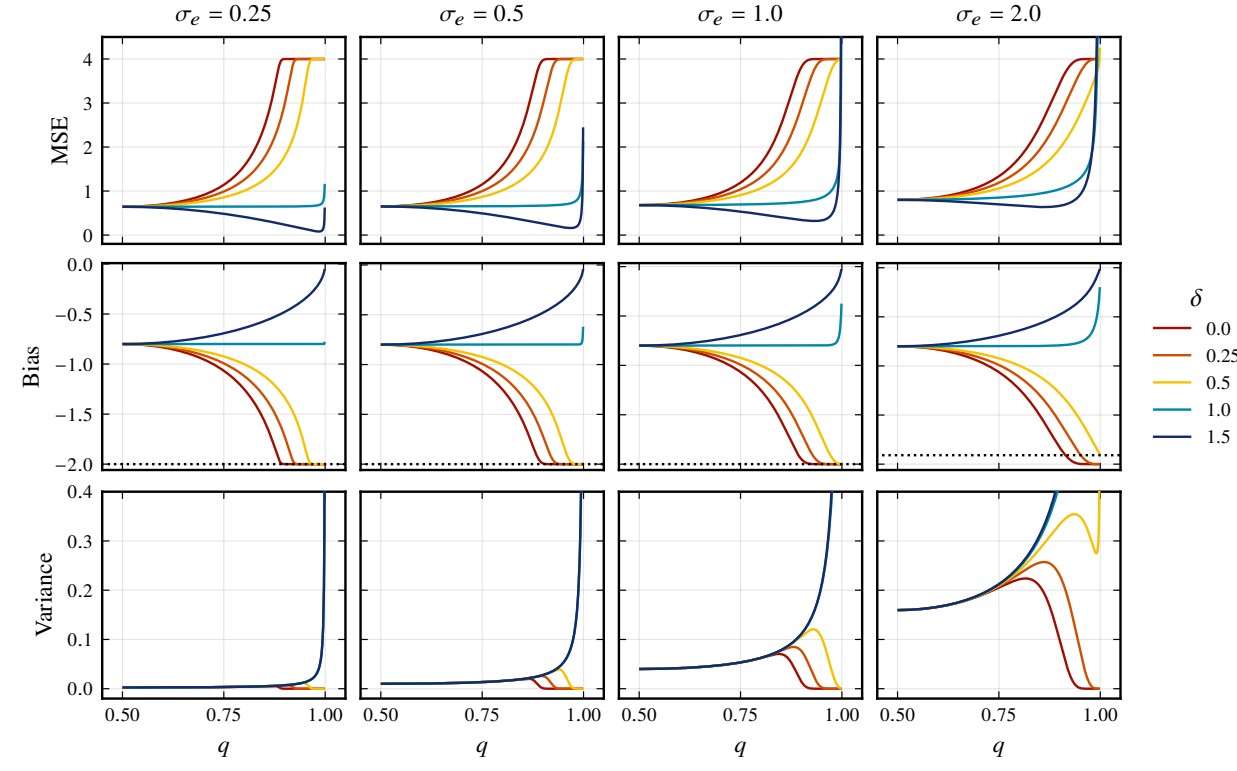

*Figure 11.* Bias, variance, and mean-squared error for a one-dimensional lasso problem, parameterized by noise level ($\sigma_\varepsilon$), class balance ($q$), and scaling ($\delta$). Dotted lines represent asymptotic bias of the lasso estimator in the case when $\delta = 1/2$.

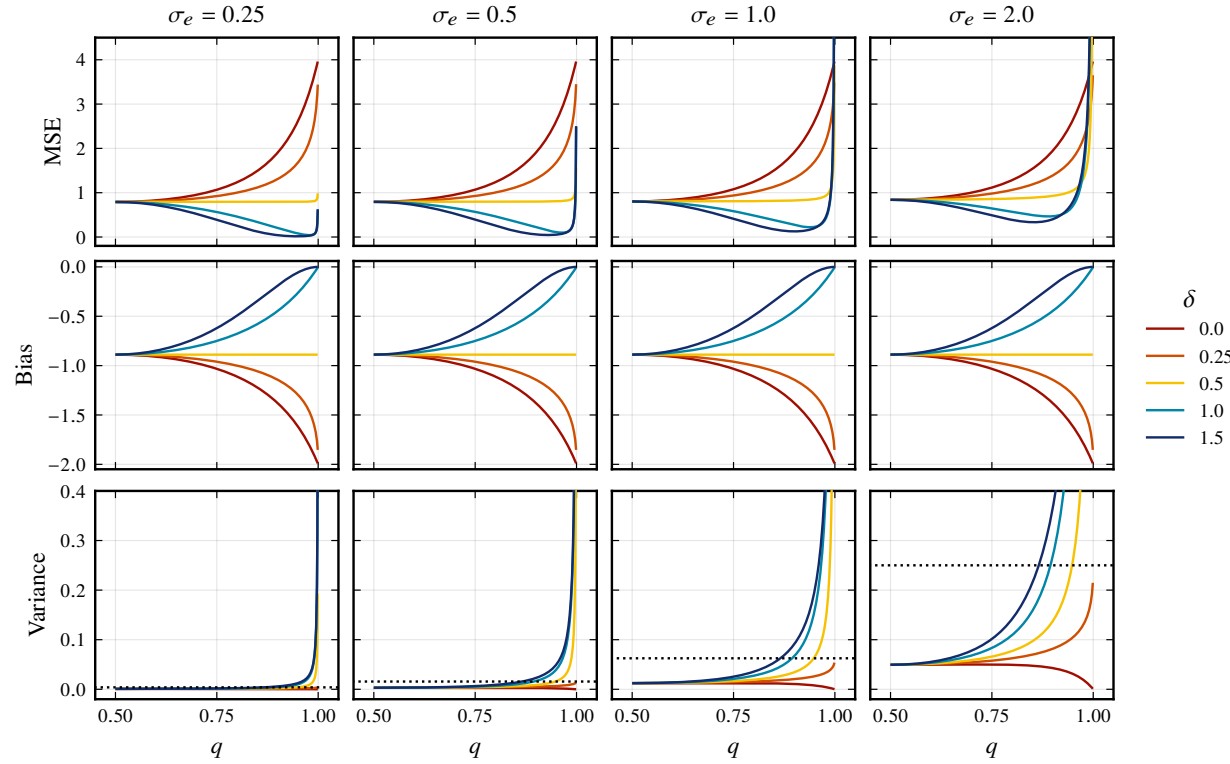

*Figure 12.* Bias, variance, and mean-squared error for one-dimensional ridge regression, parameterized by noise level ($\sigma_\varepsilon$), class balance ($q$), and scaling ($\delta$). Dotted lines represent asymptotic bias of the ridge estimator in the case of $\delta = 1/4$.

with

$$a = \frac{\beta^* \sqrt{n}}{\sigma_\varepsilon} \qquad \text{and} \qquad b = \frac{\lambda_1}{\sigma_\varepsilon \sqrt{n}}.$$

We are interested in

$$\lim_{q \to 1^+} \mathrm{E}\,\hat{\beta} = \lim_{q \to 1^+} \frac{1}{d} \left( -\theta\,\Phi\left(\frac{\theta}{\sigma}\right) - \sigma\,\phi\left(\frac{\theta}{\sigma}\right) + \gamma\,\Phi\left(\frac{\gamma}{\sigma}\right) + \sigma\,\phi\left(\frac{\gamma}{\sigma}\right) \right). \tag{14}$$

Before we proceed, note the following limits, which we will make repeated use of throughout the proof.

$$\lim_{q \to 1^+} \frac{\theta}{\sigma} = \lim_{q \to 1^+} \frac{\gamma}{\sigma} = \begin{cases} -\infty & \text{if } 0 \le \delta < \frac{1}{2}, \\ -b & \text{if } \delta = \frac{1}{2}, \\ 0 & \text{if } \delta > \frac{1}{2}, \end{cases} \tag{15}$$

Starting with the terms involving $\Phi$ inside the limit in Equation (14), for now assuming that they are well-defined and that the limits of the remaining terms also exist seperately, we have

$$\lim_{q \to 1^+} \left( -\frac{\theta}{d}\,\Phi\left(\frac{\theta}{\sigma}\right) + \frac{\gamma}{d}\,\Phi\left(\frac{\gamma}{\sigma}\right) \right) = \lim_{q \to 1^+} \left( \left( \frac{\beta^* n}{n + \lambda_2(q - q^2)^{2\delta - 1}} + \frac{\lambda_1}{n(q - q^2)^{1 - \delta} + \lambda_2(q - q^2)^\delta} \right) \Phi\left(\frac{\theta}{\sigma}\right) \right.$$

$$+ \left. \left( \frac{\beta^* n}{n + \lambda_2(q - q^2)^{2\delta - 1}} - \frac{\lambda_1}{n(q - q^2)^{1 - \delta} + \lambda_2(q - q^2)^\delta} \right) \Phi\left(\frac{\gamma}{\sigma}\right) \right)$$

$$= \lim_{q \to 1^+} \frac{\beta^* n}{n + \lambda_2(q - q^2)^{2\delta - 1}} \left( \Phi\left(\frac{\theta}{\sigma}\right) + \Phi\left(\frac{\gamma}{\sigma}\right) \right)$$

$$+ \lim_{q \to 1^+} \frac{\lambda_1}{n(q - q^2)^{1 - \delta} + \lambda_2(q - q^2)^\delta} \left( \Phi\left(\frac{\theta}{\sigma}\right) - \Phi\left(\frac{\gamma}{\sigma}\right) \right). \tag{16}$$

Considering the first term in Equation (16), we see that

$$\lim_{q \to 1^+} \frac{\beta^* n}{n + \lambda_2(q - q^2)^{2\delta - 1}} \left( \Phi\left(\frac{\theta}{\sigma}\right) + \Phi\left(\frac{\gamma}{\sigma}\right) \right) = \begin{cases} 0 & \text{if } 0 \le \delta < 1/2, \\ \frac{2n\beta^*}{n + \lambda_2}\,\Phi(-b) & \text{if } \delta = 1/2, \\ \beta^* & \text{if } \delta > 1/2. \end{cases}$$

For the second term in Equation (16), we start by observing that if $\delta = 1$, then $(q - q^2)^{\delta - 1} = 1$, and if $\delta > 1$, then $\lim_{q \to 1^+}(q - q^2)^{\delta - 1} = 0$. Moreover, the arguments of $\Phi$ approach 0 in the limit for $\delta \ge 1$, which means that the entire term vanishes in both cases ($\delta \ge 1$).

For $0 \le \delta < 1$, the limit is indeterminite of the form $\infty \times 0$. We define

$$f(q) = \Phi\left(\frac{\theta}{\sigma}\right) - \Phi\left(\frac{\gamma}{\sigma}\right) \qquad \text{and} \qquad g(q) = n(q - q^2)^{1 - \delta} + \lambda_2(q - q^2)^\delta,$$

such that we can express the limit as $\lim_{q \to 1^+} f(q)/g(q)$. The corresponding derivatives are

$$f'(q) = \left( -\frac{a}{2}(1 - 2q)(q - q^2)^{-1/2} - b(\delta - 1/2)(1 - 2q)(q - q^2)^{\delta - 3/2} \right) \phi\left(\frac{\theta}{\sigma}\right)$$

$$- \left( \frac{a}{2}(1 - 2q)(q - q^2)^{-1/2} - b(\delta - 1/2)(1 - 2q)(q - q^2)^{\delta - 3/2} \right) \phi\left(\frac{\gamma}{\sigma}\right),$$

$$g'(q) = n(1 - \delta)(1 - 2q)(q - q^2)^{-\delta} + \lambda_2\delta(1 - 2q)(q - q^2)^{\delta - 1}$$

Note that $f(q)$ and $g(q)$ are both differentiable and $g'(q) \ne 0$ everywhere in the interval $(1/2, 1)$. Now note that we have

$$\frac{f'(q)}{g'(q)} = \frac{1}{n(1 - \delta)(q - q^2)^{1/2 - \delta} + \lambda_2\delta(1 - 2q)(q - q^2)^{\delta - 1/2}}$$

$$\times \left( -\left( \frac{a}{2} + b(\delta - 1/2)(q - q^2)^{\delta - 1} \right) \phi\left(\frac{\theta}{\sigma}\right) - \left( \frac{a}{2} - b(\delta - 1/2)(q - q^2)^{\delta - 1} \right) \phi\left(\frac{\gamma}{\sigma}\right) \right). \tag{17}$$

For $0 \leq \delta < 1/2$, $\lim_{q \to 1+} f'(q)/g'(q) = 0$ since the exponential terms of $\phi$ in Equation (17) dominate in the limit.

For $\delta = 1/2$, we have

$$\lim_{q \to 1+} \frac{f'(q)}{g'(q)} = -\frac{a}{n + \lambda_2} \lim_{q \to 1+} \left( \phi\left(\frac{\theta}{\sigma}\right) + \phi\left(\frac{\gamma}{\sigma}\right) \right) = -\frac{2a\,\phi(-b)}{n + \lambda_2}$$

so that we can use L'Hôpital's rule to show that the second term in Equation (16) becomes

$$-\frac{2\beta^* \lambda_1 \sqrt{n}}{\sigma_\varepsilon (n + \lambda_2)} \phi\left(\frac{-\lambda_1}{\sigma_\varepsilon \sqrt{n}}\right). \tag{18}$$

For $\delta > 1/2$, we have

$$\lim_{q \to 1+} \frac{f'(q)}{g'(q)} = \lim_{q \to 1+} \frac{-\frac{a}{2}\left( \phi\left(\frac{\theta}{\sigma}\right) + \phi\left(\frac{\gamma}{\sigma}\right) \right)}{n(1-\delta)(q-q^2)^{1/2-\delta} + \lambda_2 \delta(1-2q)(q-q^2)^{\delta-1/2}}$$

$$+ \lim_{q \to 1+} \frac{b(\delta - 1/2)\left( \phi\left(\frac{\gamma}{\sigma}\right) - \phi\left(\frac{\theta}{\sigma}\right) \right)}{n(1-\delta)(q-q^2)^{3/2-2\delta} + \lambda_2 \delta(1-2q)(q-q^2)^{1/2}}$$

$$= 0 + \lim_{q \to 1+} \frac{b(\delta - 1/2)e^{-\frac{1}{2}\left( a^2(q-q^2)+b^2(q-q^2)^{2\delta-1} \right)} \left( e^{-ab(q-q^2)^\delta} - e^{ab(q-q^2)^\delta} \right)}{\sqrt{2\pi} \left( n(1-\delta)(q-q^2)^{3/2-2\delta} + \lambda_2 \delta(1-2q)(q-q^2)^{1/2} \right)}$$

$$= 0$$

since the exponential term in the numerator dominates.

Now we proceed to consider the terms involving $\phi$ in Equation (14). We have

$$\lim_{q \to 1+} \frac{\sigma}{d} \left( \phi\left(\frac{\gamma}{\sigma}\right) - \phi\left(\frac{\theta}{\sigma}\right) \right) = \sigma_\varepsilon \sqrt{n} \lim_{q \to 1+} \frac{\phi\left(\frac{\gamma}{\sigma}\right) - \phi\left(\frac{\theta}{\sigma}\right)}{n(q-q^2)^{1/2} + \lambda_2(q-q^2)^{2\delta-1/2}} \tag{19}$$

For $0 \leq \delta < 1/2$, we observe that the exponential terms in $\phi$ dominate in the limit, and so we can distribute the limit and consider the limits of the respective terms individually, which both vanish.

For $\delta \geq 1/2$, the limit in Equation (19) has an indeterminate form of the type $\frac{0}{0}$. Define

$$u(q) = \phi\left(\frac{\gamma}{\sigma}\right) - \phi\left(\frac{\theta}{\sigma}\right) \qquad \text{and} \qquad v(q) = n(q-q^2)^{1/2} + \lambda_2(q-q^2)^{2\delta-1/2}$$

which are both differentiable in the interval $(1/2, 1)$ and $v'(q) \neq 0$ everywhere in this interval. The derivatives are

$$u'(q) = -\phi\left(\frac{\gamma}{\sigma}\right) \frac{\gamma}{\sigma} \left( \frac{1}{2}\left( a(1-2q)(q-q^2)^{-1/2} \right) - b(\delta - 1/2)(1-2q)(q-q^2)^{\delta-3/2} \right)$$

$$+ \phi\left(\frac{\theta}{\sigma}\right) \frac{\theta}{\sigma} \left( \frac{1}{2}\left( a(1-2q)(q-q^2)^{-1/2} \right) + b(\delta - 1/2)(1-2q)(q-q^2)^{\delta-3/2} \right),$$

$$v'(q) = \frac{n}{2}(1-2q)(q-q^2)^{-1/2} + \lambda_2(2\delta - 1/2)(1-2q)(q-q^2)^{2\delta-3/2}.$$

And so

$$\frac{u'(q)}{v'(q)} = \frac{1}{n + \lambda_2(4\delta - 1)(q-q^2)^{2\delta-1}} \left( \left( a - b(2\delta - 1)(q-q^2)^{\delta-1} \right) \phi\left(\frac{\gamma}{\sigma}\right) \frac{\gamma}{\sigma} \right.$$

$$\left. + \left( a + b(2\delta - 1)(q-q^2)^{\delta-1} \right) \phi\left(\frac{\theta}{\sigma}\right) \frac{\theta}{\sigma} \right). \tag{20}$$

Taking the limit, rearranging, and assuming that the limits of the separate terms exist, we obtain

$$
\lim_{q \to 1^+} \frac{u'(q)}{v'(q)} = a \lim_{q \to 1^+} \frac{1}{n + \lambda_2(4\delta - 1)(q - q^2)^{2\delta - 1}} \left( \phi\left(\frac{\gamma}{\sigma}\right) \frac{\gamma}{\sigma} - \phi\left(\frac{\theta}{\sigma}\right) \frac{\theta}{\sigma} \right)
$$

$$
+ b(2\delta - 1) \lim_{q \to 1^+} \frac{1}{n + \lambda_2(4\delta - 1)(q - q^2)^{2\delta - 1}} \left( \phi\left(\frac{\gamma}{\sigma}\right) \left( a(q - q^2)^{\delta - 1/2} - b(q - q^2)^{2\delta - 3/2} \right) \right.
$$

$$
\left. - \phi\left(\frac{\theta}{\sigma}\right) \left( -a(q - q^2)^{\delta - 1/2} - b(q - q^2)^{2\delta - 3/2} \right) \right). \quad (21)
$$

For $\delta = 1/2$, we have

$$
\lim_{q \to 1^+} \frac{u'(q)}{v'(q)} = -\frac{a}{n + \lambda_2} \left( -b\,\phi(-b) - b\,\phi(-b) \right) + 0 = 2ab\,\phi(-b) = \frac{2\beta^* \lambda_1}{\sigma_\varepsilon^2 (n + \lambda_2)} \phi\left(\frac{-\lambda_1}{\sigma_\varepsilon \sqrt{n}}\right).
$$

Using L'Hôpital's rule, Equation (19) must consequently be

$$
\frac{2\beta^* \lambda_1 \sqrt{n}}{\sigma_\varepsilon (n + \lambda_2)} \phi\left(\frac{-\lambda_1}{\sigma_\varepsilon \sqrt{n}}\right),
$$

which cancels with Equation (18).

For $\delta > 1/2$, we first observe that the first term in Equation (21) tends to zero due to Equation (15) and the properties of the standard normal distribution. For the second term, we note that this is essentially of the same form as Equation (17) and that the limit is therefore 0 here.

**C.1. Proof of Theorem 3.2**

The variance of the elastic net estimator is given by

$$
\mathrm{Var}\,\hat{\beta}_j = \frac{1}{d^2} \left( \frac{\sigma^2}{2} \left( 2 + \mathrm{erf}\left(\frac{\theta}{\sigma\sqrt{2}}\right) - \frac{\theta}{\sigma}\sqrt{\frac{2}{\pi}} \exp\left(-\frac{\theta^2}{2\sigma^2}\right) + \mathrm{erf}\left(\frac{\gamma}{\sigma\sqrt{2}}\right) - \frac{\gamma}{\sigma}\sqrt{\frac{2}{\pi}} \exp\left(-\frac{\gamma^2}{2\gamma^2}\right) \right) \right.
$$

$$
\left. + 2\theta\sigma\,\phi\left(\frac{\theta}{\sigma}\right) + \theta^2\,\Phi\left(\frac{\theta}{\sigma}\right) + 2\gamma\sigma\,\phi\left(\frac{\gamma}{\sigma}\right) + \gamma^2\,\Phi\left(\frac{\gamma}{\sigma}\right) \right) - \left(\frac{1}{d}\,\mathrm{E}\,\hat{\beta}_j\right)^2. \quad (22)
$$

We start by noting the following identities:

$$
\theta^2 = (\beta^* n)^2 (q - q^2)^{2 - 2\delta} + \lambda_1^2 + 2\lambda_1\beta^* n(q - q^2)^{1 - \delta},
$$

$$
d^2 = n^2(q - q^2)^{2 - 2\delta} + 2n\lambda_2(q - q^2) + \lambda_2^2(q - q^2)^{2\delta},
$$

$$
\theta\sigma = -\sigma_\varepsilon \left( \beta^* n^{3/2}(q - q^2)^{3/2 - 2\delta} + \sqrt{n}\lambda_1(q - q^2)^{1/2 - \delta} \right),
$$

$$
\frac{\theta^2}{\sigma^2} = a^2(q - q^2) + b^2(q - q^2)^{2\delta - 1} + 2ab(q - q^2)^\delta,
$$

$$
\frac{\sigma}{d} = \frac{\sigma_\varepsilon \sqrt{n}}{n(q - q^2)^{\frac{1}{2}} + \lambda_2(q - q^2)^{2\delta - 1/2}}.
$$

Expansions involving $\gamma$ instead of $\theta$ have identical expansions up to sign changes of the individual terms. Also recall the definitions provided in the proof of Theorem 3.1.

Starting with the case when $0 \le \delta < 1/2$, we write the limit of Equation (22) as

$$\lim_{q \to 1^+} \operatorname{Var} \hat{\beta}_j$$

$$= \sigma_\varepsilon^2 n \lim_{q \to 1^+} \frac{1}{\left(n(q-q^2)^{1/2} + \lambda_2(q-q^2)^{2\delta-1/2}\right)^2} \left(1 + \operatorname{erf}\left(\frac{\theta}{\sigma\sqrt{2}}\right) - \frac{\theta}{\sigma}\sqrt{\frac{2}{\pi}} \exp\left(-\frac{\theta^2}{2\sigma^2}\right)\right)$$

$$+ \sigma_\varepsilon^2 n \lim_{q \to 1^+} \frac{1}{\left(n(q-q^2)^{1/2} + \lambda_2(q-q^2)^{2\delta-1/2}\right)^2} \left(1 + \operatorname{erf}\left(\frac{\gamma}{\sigma\sqrt{2}}\right) - \frac{\gamma}{\sigma}\sqrt{\frac{2}{\pi}} \exp\left(-\frac{\gamma^2}{2\sigma^2}\right)\right)$$

$$+ \lim_{q \to 1^+} \frac{2\theta\sigma}{d^2} \phi\left(\frac{\theta}{\sigma}\right) + \lim_{q \to 1^+} \frac{\theta^2}{d^2} \Phi\left(\frac{\theta}{\sigma}\right) + \lim_{q \to 1^+} \frac{2\gamma}{d^2} \sigma \phi\left(\frac{\gamma}{\sigma}\right) + \lim_{q \to 1^+} \frac{\gamma^2}{d^2} \Phi\left(\frac{\gamma}{\sigma}\right)$$

$$- \left(\lim_{q \to 1^+} \frac{1}{d} \operatorname{E} \hat{\beta}_j\right)^2,$$

assuming, for now, that all limits exist. Next, let

$$f_1(q) = 1 + \operatorname{erf}\left(\frac{\theta}{\sigma\sqrt{2}}\right) - \frac{\theta}{\sigma}\sqrt{\frac{2}{\pi}} \exp\left(-\frac{\theta^2}{2\sigma^2}\right),$$

$$f_2(q) = 1 + \operatorname{erf}\left(\frac{\gamma}{\sigma\sqrt{2}}\right) - \frac{\gamma}{\sigma}\sqrt{\frac{2}{\pi}} \exp\left(-\frac{\gamma^2}{2\sigma^2}\right),$$

$$g(q) = \left(n^2(q-q^2) + 2n\lambda_2(q-q^2)^{2\delta} + \lambda_2^2(q-q^2)^{4\delta-1}\right)^2.$$

And

$$f_1'(q) = \frac{\theta^2}{\sigma^2}\sqrt{\frac{2}{\pi}} \exp\left(-\frac{\theta^2}{2\sigma^2}\right),$$

$$f_2'(q) = \frac{\gamma^2}{\sigma^2}\sqrt{\frac{2}{\pi}} \exp\left(-\frac{\gamma^2}{2\sigma^2}\right),$$

$$g'(q) = (1 - 2q)\left((q-q^2)^{-1} + 4n\delta\lambda_2(q-q^2)^{2\delta-1} + \lambda_2^2(4\delta-1)(q-q^2)^{4\delta-2}\right).$$

$f_1$, $f_1$ and $g$ are differentiable in $(1/2, 1)$ and $g'(q) \ne 0$ everywhere in this interval. $f_1/g$ and $f_2/g$ are indeterminate of the form $0/0$. And we see that

$$\lim_{q \to 1^+} \frac{f_1'(q)}{g'(q)} = \lim_{q \to 1^+} \frac{f_2'(q)}{g'(q)} = 0$$

due to the dominance of the exponential terms as $\theta/\sigma$ and $\gamma/\sigma$ both tend to $-\infty$. Thus $f_1/g$ and $f_2/g$ also tend to 0 by L'Hôpital's rule. Similar reasoning shows that

$$\lim_{q \to 1^+} \frac{2\theta\sigma}{d^2} \phi\left(\frac{\theta}{\sigma}\right) = \lim_{q \to 1^+} \frac{\theta^2}{d^2} \Phi\left(\frac{\theta}{\sigma}\right) = 0.$$

The same result applies to the respective terms involving $\gamma$. And since we in Theorem 3.1 showed that $\lim_{q \to 1^+} \frac{1}{d} \operatorname{E} \hat{\beta}_j = 0$, the limit of Equation (22) must be 0.

For $\delta = 1/2$, we start by establishing that

$$\lim_{q \to 1^+} \int_{-\infty}^{-\lambda} (z+\lambda)^2 f_Z(z) \, dz = \lim_{q \to 1^+} \left(\sigma^2 \int_{-\infty}^{\frac{\theta}{\sigma}} y^2 \phi(y) \, dy + 2\theta\sigma \int_{-\infty}^{\frac{\theta}{\sigma}} y \phi(y) \, dy + \theta^2 \int_{-\infty}^{\frac{\theta}{\sigma}} \phi(y) \, dy\right) \quad (23)$$

is a positive constant since $\theta/\sigma \to -b$, $\sigma = \sigma_\varepsilon \sqrt{n}$, $\theta \to -\lambda$, and $\theta\sigma \to -\sigma_\varepsilon \sqrt{n}\lambda$. An identical argument can be made in the case of $\lim_{q \to 1^+} \int_\lambda^\infty (z-\lambda)^2 f_Z(z) \, dz$. We then have

$$\lim_{q \to 1^+} \frac{1}{d^2} \int_{-\infty}^{-\lambda} (z+\lambda)^2 f_Z(z) \, dz = \frac{C^+}{\lim_{q \to 1^+} d^2} = \frac{C^+}{0} = \infty,$$

where $C^+$ is some positive constant. And because $\lim_{q \to 1^+} \frac{1}{d} \operatorname{E} \hat{\beta}_j = \beta^*$ (Theorem 3.1), the limit of Equation (22) must be $\infty$.

Finally, for the case when $\delta > 1/2$, we have

$$
\lim_{q \to 1^+} \frac{1}{d^2} \left( \sigma^2 \int_{-\infty}^{\frac{\theta}{\sigma}} y^2 \, \phi(y) \, \mathrm{d}y + 2\theta\sigma \int_{-\infty}^{\frac{\theta}{\sigma}} y \, \phi(y) \, \mathrm{d}y + \theta^2 \int_{-\infty}^{\frac{\theta}{\sigma}} \phi(y) \, \mathrm{d}y \right)
$$

$$
= \lim_{q \to 1^+} \left( \frac{n\sigma^2}{\left( n(q-q^2)^{1/2} + \lambda_2(q-q^2)^{2\delta-1/2} \right)^2} \int_{-\infty}^{\frac{\theta}{\sigma}} y^2 \, \phi(y) \, \mathrm{d}y \right.
$$

$$
- \frac{2\sigma_\varepsilon \sqrt{n} \left( \beta^* n(q-q^2)^{1-\delta} - \lambda_1 \right)}{\left( n(q-q^2)^{3/4-\delta/2} + \lambda_2(q-q^2)^{3\delta/2-1/4} \right)^2} \int_{-\infty}^{\frac{\theta}{\sigma}} y \, \phi(y) \, \mathrm{d}y
$$

$$
\left. + \left( \frac{-\beta^* n(q-q^2)^{1-\delta} - \lambda_1}{n(q-q^2)^{1-\delta} + \lambda_2(q-q^2)^{\delta}} \right)^2 \int_{-\infty}^{\frac{\theta}{\sigma}} \phi(y) \, \mathrm{d}y \right). \quad (24)
$$

Inspection of the exponents involving the factor $(q - q^2)$ shows that the first term inside the limit will dominate. And since the upper limit of the integrals, $\theta/\sigma \to 0$ as $q \to 1^+$, the limit must be $\infty$.

## C.2. Proof of Corollary A.2

We have

$$
\lim_{q \to 1^+} \operatorname{Var} \hat{\beta}_j = \lim_{q \to 1^+} \frac{\sigma^2}{d^2} \left( \frac{\sigma_\varepsilon \sqrt{n}(q-q^2)^{1/2-\delta}}{n(q-q^2)^{1-\delta} + \lambda_2(q-q^2)^{\delta}} \right)^2 = \frac{\sigma_\varepsilon^2 n}{\lambda_2^2} \lim_{q \to 1^+} (q-q^2)^{1-4\delta},
$$

from which the result follows directly.

## C.3. Proof of Theorem 3.3

### C.3.1. EXPECTED VALUE

Starting with the expected value, our proof follows a similar structure as in the proof for Theorem 3.1 (Appendix C). We start by noting the values of some of the important terms. As before we will drop the subscript $j$ everywhere to simplify notation. We have

$$
\mu = \beta^* (q - q^2)^\omega, \qquad\qquad \frac{\theta}{\sigma} = -a\sqrt{q - q^2} - b(q - q^2)^{\omega - 1/2},
$$

$$
\sigma = \sigma_\varepsilon \sqrt{n(q - q^2)}, \qquad\qquad \frac{\gamma}{\sigma} = a\sqrt{q - q^2} - b(q - q^2)^{\omega - 1/2},
$$

$$
d = n(q - q^2) + \lambda_2(q - q^2)^\omega. \qquad\qquad \frac{\theta}{d} = \frac{-\beta^* n - \lambda_1(q - q^2)^{\omega - 1}}{n + \lambda_2(q - q^2)^{\omega - 1}},
$$

$$
\theta = -\beta^* n(q - q^2) - \lambda_1(q - q^2)^\omega, \qquad\qquad \frac{\gamma}{d} = \frac{\beta^* n - \lambda_1(q - q^2)^{\omega - 1}}{n + \lambda_2(q - q^2)^{\omega - 1}},
$$

$$
\gamma = \beta^* n(q - q^2) - \lambda_1(q - q^2)^\omega.
$$

First note the following limit (which is analogous to that in Equation (15)).

$$
\lim_{q \to 1^+} \frac{\theta}{\sigma} = \lim_{q \to 1^+} \frac{\gamma}{\sigma} = \begin{cases} -\infty & \text{if } 0 \leq \omega < \frac{1}{2}, \\ -b & \text{if } \omega = \frac{1}{2}, \\ 0 & \text{if } \omega > \frac{1}{2}. \end{cases} \quad (25)
$$

As in Appendix C, we are looking to compute the following limit:

$$
\lim_{q \to 1^+} \operatorname{E} \hat{\beta} = \lim_{q \to 1^+} \frac{1}{d} \left( -\theta \, \Phi\left(\frac{\theta}{\sigma}\right) - \sigma \, \phi\left(\frac{\theta}{\sigma}\right) + \gamma \, \Phi\left(\frac{\gamma}{\sigma}\right) + \sigma \, \phi\left(\frac{\gamma}{\sigma}\right) \right). \quad (26)
$$

Starting with the terms involving $\Phi$ and assuming that the limit can be distributed, we have

$$\lim_{q \to 1^+} \left( -\frac{\theta}{d} \Phi\left(\frac{\theta}{\sigma}\right) + \frac{\gamma}{d} \Phi\left(\frac{\gamma}{\sigma}\right) \right) = \lim_{q \to 1^+} \frac{\beta^* n + \lambda_1 (q - q^2)^{\omega - 1}}{n + \lambda_2 (q - q^2)^{\omega - 1}} \Phi\left(\frac{\theta}{\sigma}\right)$$

$$+ \lim_{q \to 1^+} \frac{\beta^* n - \lambda_1 (q - q^2)^{\omega - 1}}{n + \lambda_2 (q - q^2)^{\omega - 1}} \Phi\left(\frac{\gamma}{\sigma}\right)$$

$$= \begin{cases} 0 & \text{if } 0 \le \omega < 1, \\ \frac{\beta^* n}{n + \lambda_2} & \text{if } \omega = 1, \\ \beta^* & \text{if } \omega > 1. \end{cases} \tag{27}$$

The derivation of the first case in Equation (27) depends on $\omega$. For $0 \le \omega \le 1/2$, it stems from the facts that $\Phi(\theta/\sigma) \to 0$ and $\Phi(\theta/\sigma) \to 0$ as $q \to 1^+$ together with the existence of the $(q - q^2)^{\omega - 1}$ factor in both numerator and denominator. For $1/2 \le \omega < 1$, the terms cancel each other out. In the second case, when $\omega = 1$, the result stems from $\Phi(\theta/\sigma)$ and $\Phi(\gamma/\sigma)$ both tending to 1/2 as $q \to 1^+$. And finally for $\omega > 1$, the terms involving the $(q - q^2)^{\omega - 1}$ factors vanish and again the values of the cumulative distribution functions tend to 1/2.

Now, we turn to the terms involving the probability density function $\phi$. Again, we assume the limit is distributive so that

$$\lim_{q \to 1^+} \frac{\sigma}{d} \left( \phi\left(\frac{\gamma}{\sigma}\right) - \left(\frac{\theta}{\sigma}\right) \right) = \lim_{q \to 1^+} \frac{\sigma}{d} \phi\left(\frac{\gamma}{\sigma}\right) - \lim_{q \to 1^+} \frac{\sigma}{d} \phi\left(\frac{\theta}{\sigma}\right). \tag{28}$$

Starting with the first term on the right-hand side of Equation (28), we have

$$\lim_{q \to 1^+} \frac{\sigma}{d} \phi\left(\frac{\gamma}{\sigma}\right) = \frac{\sigma_\varepsilon \sqrt{n}\, \phi\left(\frac{\gamma}{\sigma}\right)}{n(q - q^2)^{1/2} + \lambda_2 (q - q^2)^{\omega - 1/2}}.$$

For $0 \le \omega < 1/2$, this limit is 0 since the exponential terms in the numerator will dominate as $q \to 1^+$. For $\omega = 1/2$, we have the limit $\sigma_\varepsilon \sqrt{n}\, \phi(-b)/\lambda_2$. For $\omega > 1/2$, the limit is indeterminate of the type $0/0$. Let

$$f_1(q) = \phi\left(\frac{\gamma}{\sigma}\right) \qquad \text{and} \qquad g(q) = n(q - q^2)^{1/2} + \lambda_2 (q - q^2)^{\omega - 1/2}$$

and observe that $f_1$ and $g$ are differentiable and $g'(q) \ne 0$ for $q \in (1/2, 1)$. The derivatives are

$$f_1'(q) = -\left( \frac{a}{2}(1 - 2q)(q - q^2)^{-1/2} - b(\omega - 1/2)(1 - 2q)(q - q^2)^{\omega - 3/2} \right) \frac{\gamma}{\sigma} \phi\left(\frac{\gamma}{\sigma}\right),$$

$$g'(q) = \frac{n}{2}(1 - 2q)(q - q^2)^{-1/2} + \lambda_2(\omega - 1/2)(1 - 2q)(q - q^2)^{\omega - 3/2}.$$

Next, we find that

$$\frac{f_1'(q)}{g'(q)} = \frac{-a + b(2\omega - 1)(q - q^2)^{\omega - 1}}{n + \lambda_2(2\omega - 1)(q - q^2)^{\omega - 1}} \left(\frac{\gamma}{\sigma}\right) \phi\left(\frac{\gamma}{\sigma}\right). \tag{29}$$

Taking the limit of Equation (29) and invoking L'Hôpital's rule yields

$$\lim_{q \to 1^+} \frac{f_1'(q)}{g'(q)} = 0$$

both when $1/2 < \omega < 1$ and $\omega \ge 1$ since $\gamma/\sigma$ tends to 0 as $q \to 1^+$ for $\omega > 1/2$ and the $\phi$ term tends to a constant, plus the fact that the remaining factor in the expression also tends to a constant since the terms involving $(q - q^2)^{\omega - 1}$ vanish when $\omega > 1$, are constant when $\omega = 1$, and cancel each other out in the limit when $\omega < 1$.

Finally, if we now consider the second term on the right-hand side of Equation (28), set $f_2(q) = \phi(\theta/\sigma)$, and perform the same steps as above, we find that the limits are the same in all cases, which means that the limits in Equation (28) cancel in the case when $\omega = 1/2$ and therefore that

$$\lim_{q \to 1^+} \frac{\sigma}{d} \left( \phi\left(\frac{\gamma}{\sigma}\right) - \left(\frac{\theta}{\sigma}\right) \right) = 0$$

for $0 \le \omega$. The limit in Equation (26) is given by Equation (27).

### C.3.2. VARIANCE

The proof for the variance result is in many ways equivalent to that in the case of variance of the normalized unweighted elastic net (Appendix C.1) and we therefore omit many of the details here.

In the case of $0 \leq \omega < 1/2$, the proof is simplified since the $d^2$ term tends to $\infty$ whilst the numerator takes the same limit as in the normalized case, which means that the limit is 0 in this case. For $\omega = 1/2$, we consider Equation (23) and observe that it again tends to a positive constant whilst $\lim_{q \to 1^+} d^2 = 0$, which means that the limit of the expression, and hence variance of the estimator, tends to $\infty$. For $\omega > 1/2$, an identical argument for the expression in Equation (24) holds and the limit is therefore $\infty$ in this case as well.

## C.4. Proof of Theorem A.1

If $X_i \sim \mathrm{Normal}(\mu, \sigma)$, then $|X_i| \sim \mathrm{FoldedNormal}(\mu, \sigma)$. By the Fisher–Tippett–Gnedenko theorem, we know that $(\max_i |X_i| - b_n)/a_n$ converges in distribution to either the Gumbel, Fréchet, or Weibull distribution, given a proper choice of $a_n > 0$ and $b_n \in \mathbb{R}$. A sufficient condition for convergence to the Gumbel distribution for a absolutely continuous cumulative distribution function (Nagaraja & David, 2003, Theorem 10.5.2) is

$$\lim_{x \to \infty} \frac{d}{dx} \left( \frac{1 - F(x)}{f(x)} \right) = 0.$$

We have

$$
\begin{aligned}
\frac{1 - F_Y(x)}{f_Y(x)} &= \frac{1 - \frac{1}{2}\operatorname{erf}\left(\frac{x-\mu}{\sqrt{2\sigma^2}}\right) - \frac{1}{2}\operatorname{erf}\left(\frac{x+\mu}{\sqrt{2\sigma^2}}\right)}{\frac{1}{\sqrt{2\pi\sigma^2}}e^{\frac{-(x-\mu)^2}{2\sigma^2}} + \frac{1}{\sqrt{2\pi\sigma^2}}e^{\frac{-(x+\mu)^2}{2\sigma^2}}} \\
&= \frac{2 - \Phi\left(\frac{x-\mu}{\sigma}\right) - \Phi\left(\frac{x+\mu}{\sigma}\right)}{\frac{1}{\sigma}\left(\phi\left(\frac{x-\mu}{\sigma}\right) + \phi\left(\frac{x+\mu}{\sigma}\right)\right)} \\
&\to \frac{\sigma(1 - \Phi(x))}{\phi(x)} \text{ as } n \to n,
\end{aligned}
$$

where $\phi$ and $\Phi$ are the probability distribution and cumulative density functions of the standard normal distribution respectively. Next, we follow Nagaraja & David (2003, example 10.5.3) and observe that

$$\frac{d}{dx}\frac{\sigma(1 - \Phi(x))}{\phi(x)} = \frac{\sigma x(1 - \Phi(x))}{\phi(x)} - \sigma \to 0 \text{ as } x \to \infty$$

since

$$\frac{1 - \Phi(x)}{\phi(x)} \sim \frac{1}{x}.$$

In this case, we may take $b_n = F_Y^{-1}(1 - 1/n)$ and $a_n = \left(nf_Y(b_n)\right)^{-1}$.

# D. Additional Experiments

In this section we present additional and extended results from the main section.

## D.1. Extended Example of Regularization Paths

In Figure 13 we show an extended example of lasso paths for different real data sets and types of normalization.

## D.2. Power and False Discoveries for Multiple Features

Here, we study how the power of correctly detecting $k = 10$ signals under $q_j$ linearly spaced in $[0.5, 0.99]$ (Figure 14(a)). We set $\beta_j^* = 2$ for each of the signals, use $n = 100\,000$, and let $\sigma_\varepsilon = 1$. The level of regularization is set to $\lambda_1 = n4^\delta/10$. As we can see, the power is directly related to $q_j$ and for unbalanced features stronger the higher the choice of $\delta$ is.

We also consider a version of the same setup, but with $p$ linearly spaced in $[20, 100]$ and compute normalized mean-squared error (NMSE) and false discovery rate (FDR) (Figure 14(b)). As before, we let $k = 10$ and consider three different levels of

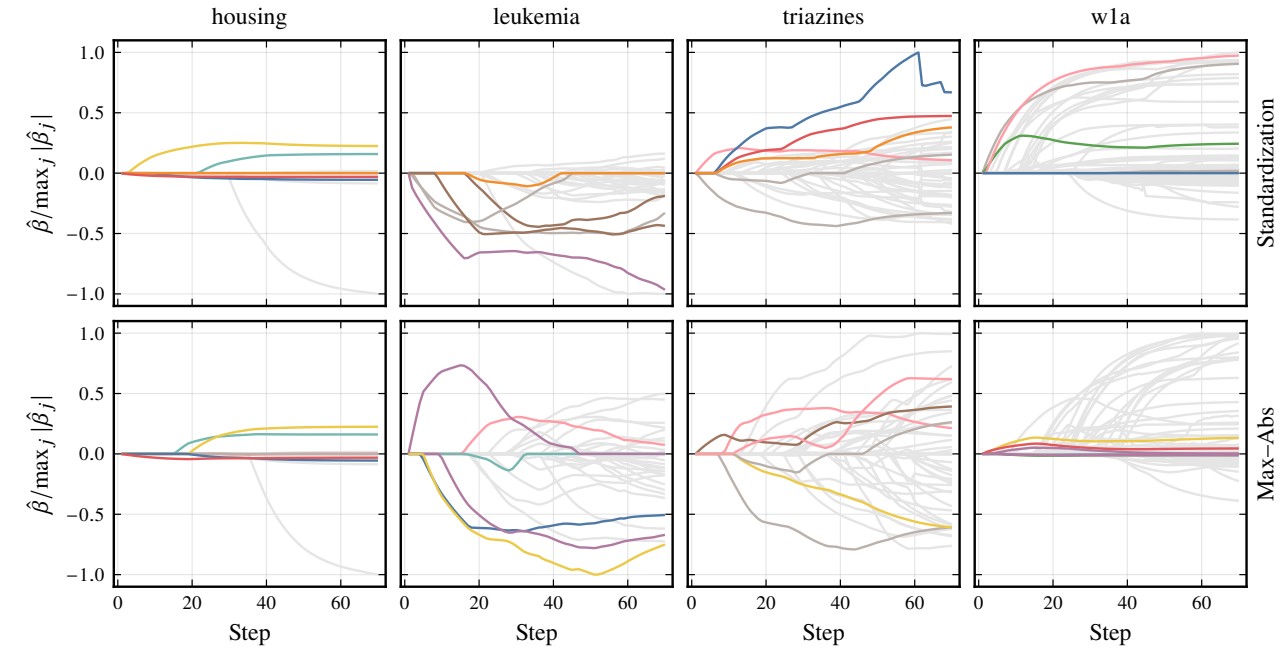

*Figure 13.* Lasso paths for real datasets using two types of normalization: standardization and maximum absolute value normalization (max–abs). We have fit the lasso path to two different datasets: `housing` (Harrison & Rubinfeld, 1978), `leukemia` (Golub et al., 1999), `triazines` (King, 2024), and `w1a` (Platt, 1998). (See Appendix E for more information about these data sets.) For each dataset, we have colored the coefficients if they were among the first five to become non-zero under either of the two normalization schemes. We see that the paths differ with regards to the size as well as the signs of the coefficients, and that, in addition, the features to become active first differ between the normalization types.

class imbalance. The remaining $p - k$ features have class balances spaced evenly on a logarithmic scale from 0.5 to 0.99. Unsurprisingly, the increase in power gained from selecting $\delta = 1$ imposes increased false discovery rates. We also see that the mean-squared error depends on class balance. In line with our previous results, $\delta \in \{0, 1/2\}$ appears to work well for balanced features whilst $\delta = 1$ works better when there are large imbalances. In the case when $q_j = 0.99$, the model under scaling with $\delta = 0$ does not detect any of the true signals.

The results (Figure 4, and Figure 15 in Appendix A.6) show that class balance has considerable effect, particularly in the case of no scaling ($\delta = 0$), which corroborates our theory from Section 3.1. At $q_j = 0.99$, for instance, the estimate ($\hat{\beta}_{20}$) is consistently zero when $\delta = 0$. For $\delta = 1$, we see that class imbalance increases the variance of the estimates. What is also clear is that the variance of the estimates increase with class imbalance and that this effect increases together with $\delta$.

The level of correlation between the features introduces additional variance in the estimates but also seems to increase the effect of class imbalance in the cases when $\delta = 0$ or $1/2$.

### D.2.1. PREDICTIVE PERFORMANCE FOR SIMULATED DATA

In this experiment, we consider predictive performance in terms of mean-squared error of the lasso and ridge regression given different levels of class balance ($q_j \in \{0.5, 0.9, 0.99\}$), signal-to-noise ratio, and normalization ($\delta$). All of the features are binary, but here we have used $n = 300$ and $p = 1000$. The $k = 10$ first features correspond to true signals with $\beta_j^* = 1$ and all have class balance $q$. To set signal-to-noise ratio levels, we rely on the same choice as in Hastie et al. (2020) and use a log-spaced sequence of values from 0.05 to 6. We use standard hold-out validation with equal splits for training, validation, and test sets. And we fit a full lasso path, parameterized by a log-spaced grid of 100 values[4], from $\lambda_{\max}$ (the value of $\lambda$ at which the first feature enters the model) to $10^{-2}\lambda_{\max}$ on the training set and pick $\lambda$ based on validation set error. Then we compute the hold-out test set error and aggregate the results across 100 iterations.

The results (Figure 17) show that the optimal normalization type in terms of prediction power depends on the class balance

---

[4]This is a standard choice of grid, used for instance by Friedman et al. (2010)

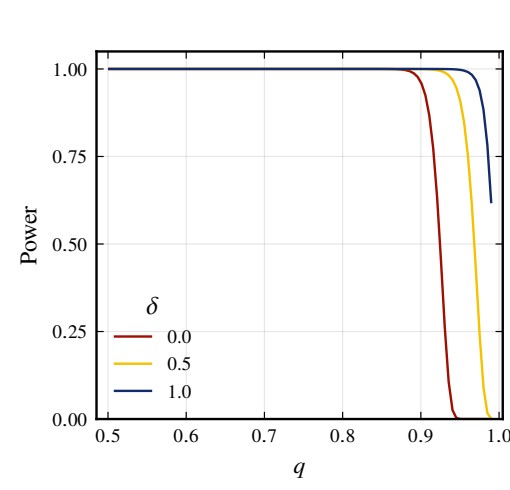

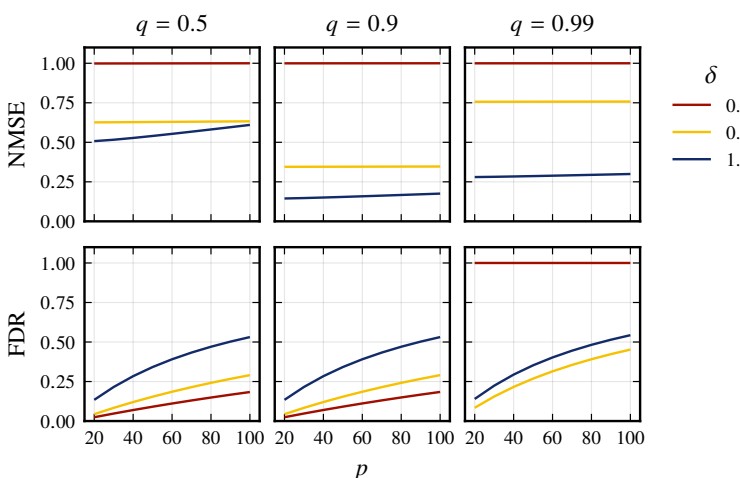

(a) The power (probability of detecting all true signals) of the lasso. In our orthogonal setting, power is constant over $p$, which is why we have omitted the parameter in the plot.

(b) NMSE and FDR: the rate of coefficients incorrectly set to non-zero (false discoveries) to the total number of estimated coefficients that are nonzero (discoveries).

*Figure 14.* Normalized mean-squared error (NMSE), false discovery rate (FDR), and power for a lasso problem with $k = 10$ true signals (nonzero $\beta_j^*$), varying $p$, and $q_j \in [0.5, 0.99]$. The noise level is set at $\sigma_\varepsilon = 1$ and $\lambda_1 = 0.02$.

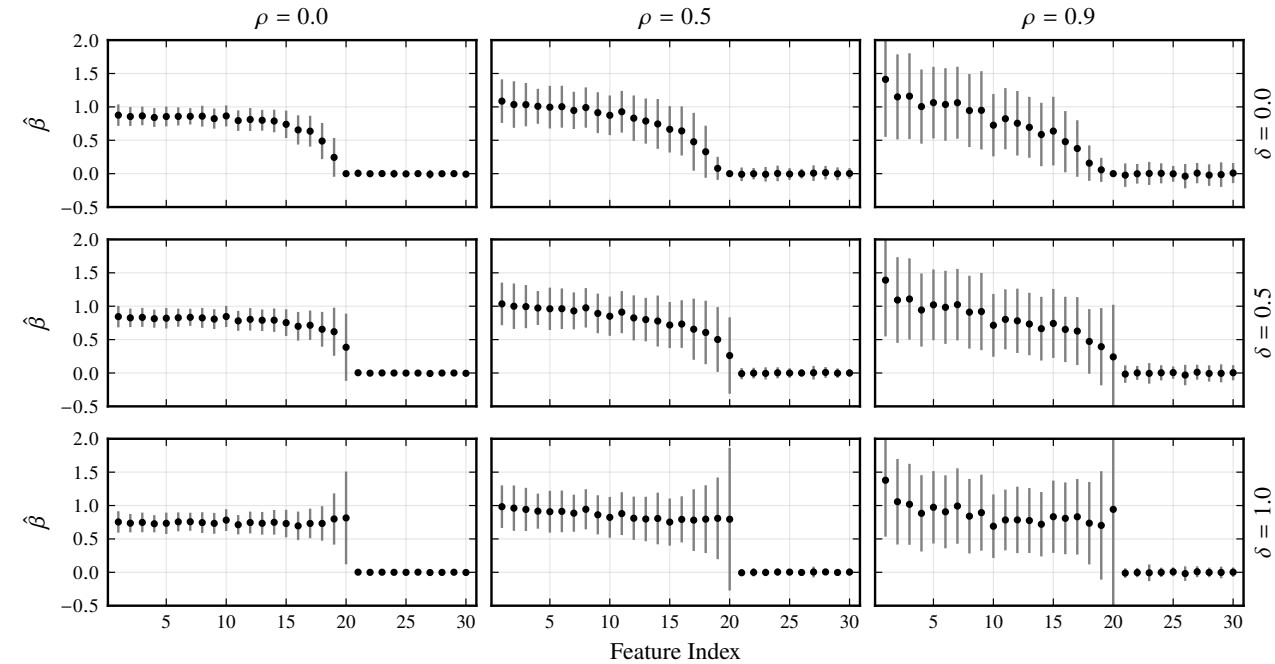

*Figure 15.* Estimates of the regression coefficients from the lasso, $\hat{\boldsymbol{\beta}}$, for the first 30 coefficients in the experiment. All of the features are binary and the first 20 features correspond to true signals with $\beta_j^* = 2$ and geometrically decreasing class balance from 0.5 to 0.99. The remaining features have class balance $q_j \in [0.5, 0.99]$ distributed linearly among the features. The plot shows means and standard deviations averaged over 100 iterations.

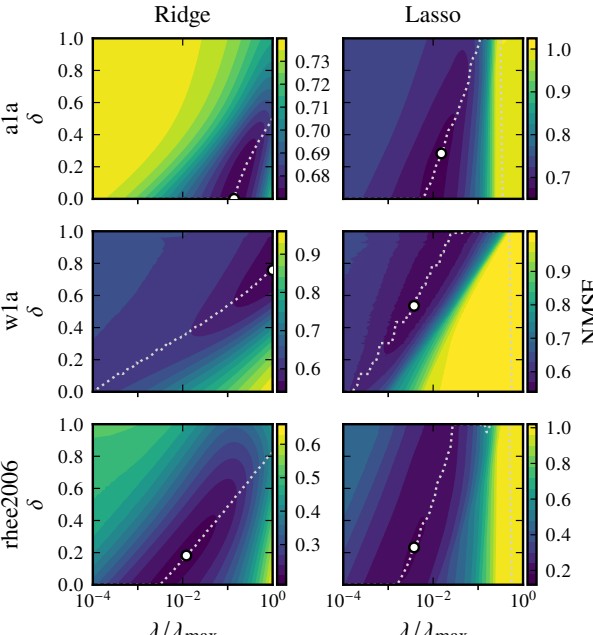

*Figure 16.* Contour plots of normalized mean-squared error (NMSE) for the hold-out validation set across a grid of $\delta$ and $\lambda$ values for ridge regression and the lasso. The dotted path shows the smallest NMSE as a function of $\lambda$. The dot marks the combination with the smallest error.

of the true signals. If the imbalance is severe, then we gain by using $\delta = 1/2$ or 1, which gives a chance of recovering the true signals. If everything is balanced, however, then we do better by not scaling. In general, $\delta = 1/2$ works well for these specific combinations of settings.

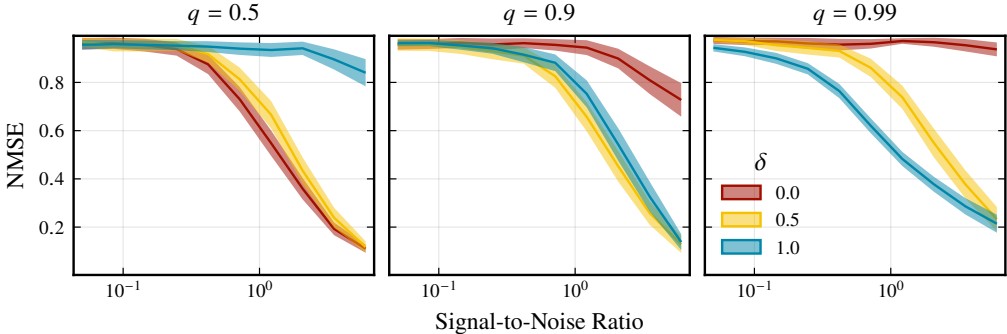

*Figure 17.* Normalized mean-squared prediction error in a lasso model for different types of normalization ($\delta$), types of class imbalances ($q_j$), and signal-to-noise ratios (0.05 to 6) in a data set with $n = 300$ observations and $p = 1000$ features. The error is aggregated test-set error from hold-out validation with 100 observations in each of the training, validation, and test sets. The plot shows means and Student's $t$-based 95% confidence intervals.

In Figure 18, we have, in addition to NMSE on the validation set, also plotted the size of the support of the lasso (cardinality of the set of features that have corresponding nonzero coefficients). Here we only show results for $\delta \in \{0, 1/2, 1\}$. It is clear that $\delta = 1/2$ works quite well for all of these three data sets, being able to attain a value close to the mininum for each of the three data sets. This is not the case for $\delta \in \{0, 1\}$, for which the best possible prediction error is considerably worse. This is particularly the case with $\delta = 0$ and the w1a data set. The dependency between $\lambda$ and $\delta$ is also visible here by looking at the support size.

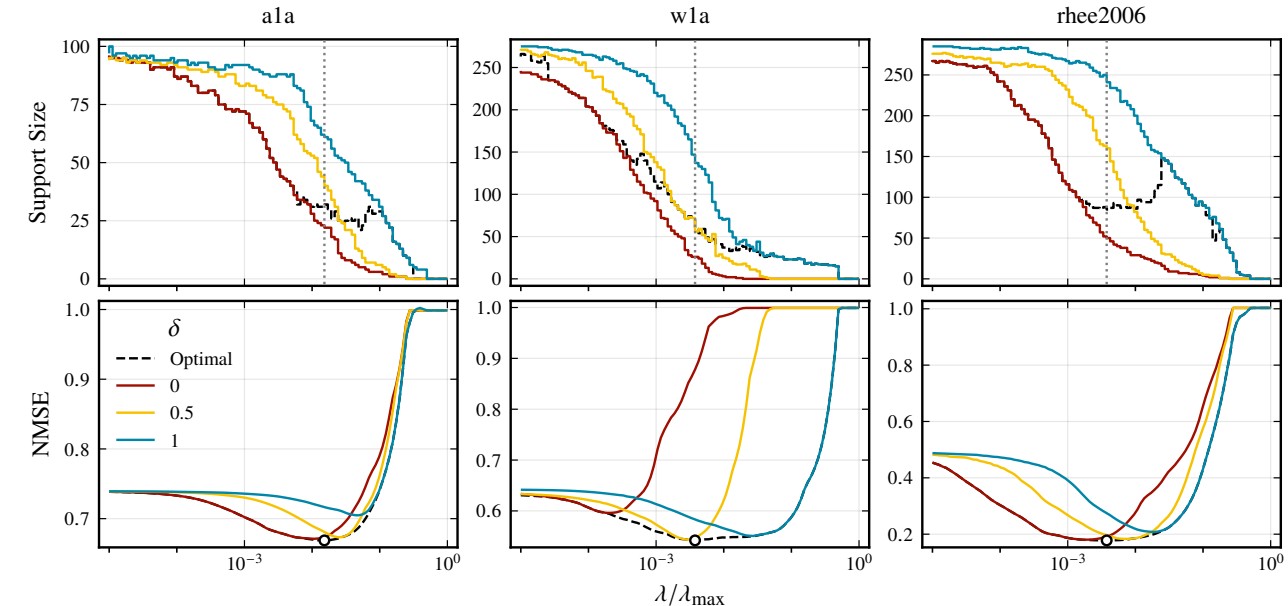

*Figure 18.* Support size and normalized mean-squared error (NMSE) for the validation set for the lasso fit to datasets `a1a`, `w1a`, and `rhee2006` across combinations of $\delta$ and $\lambda$. The optimal $\delta$ is marked with dashed black lines and the best combination of $\delta$ (among 0, 1/2, and 1) and $\lambda$ is shown as a dot.

*Table 2.* Details of the real datasets used in the experiments, The median $q$ value refers to the median of the proportion of ones for the binary features in the data. Note that in the case of `housing`, there is only a single binary feature.

| Dataset | $n$ | $p$ | Response | Design | Median $q$ |
|---|---|---|---|---|---|
| a1a | 1605 | 123 | binary | binary | 0.970 |
| w1a | 2477 | 300 | binary | binary | 0.976 |
| rhee2006 | 842 | 361 | continuous | binary | 0.995 |
| housing | 506 | 13 | continuous | mixed | 0.931 |
| leukemia | 38 | 7129 | binary | continuous | |
| triazines | 186 | 60 | continuous | mixed | 0.973 |

### D.3. Interactions

In Figure 19 we show an version of the result in Figure 7 with different types of signals. Irrespective of the signal, strategy 2 still performs best.

### D.4. The Weighted Elastic Net

In Figure 20, we show a version of Figure 8 with various settings for $\alpha$ (the balance between the lasso and ridge penalties). Our previous conclusions do not seem to be affected by this choice, but as expected the class balance bias decreases as $\alpha$ approaches 0 (ridge regression).

## E. Summary of Data Sets

In Table 2 we summarize the data sets we use in our paper.

We also visualize the distribution of class balance among all the binary features in Figure 21. We note that the clas imbalance for these data sets is quite severe, which is common for data sets, particularly in the high-dimensional regime.

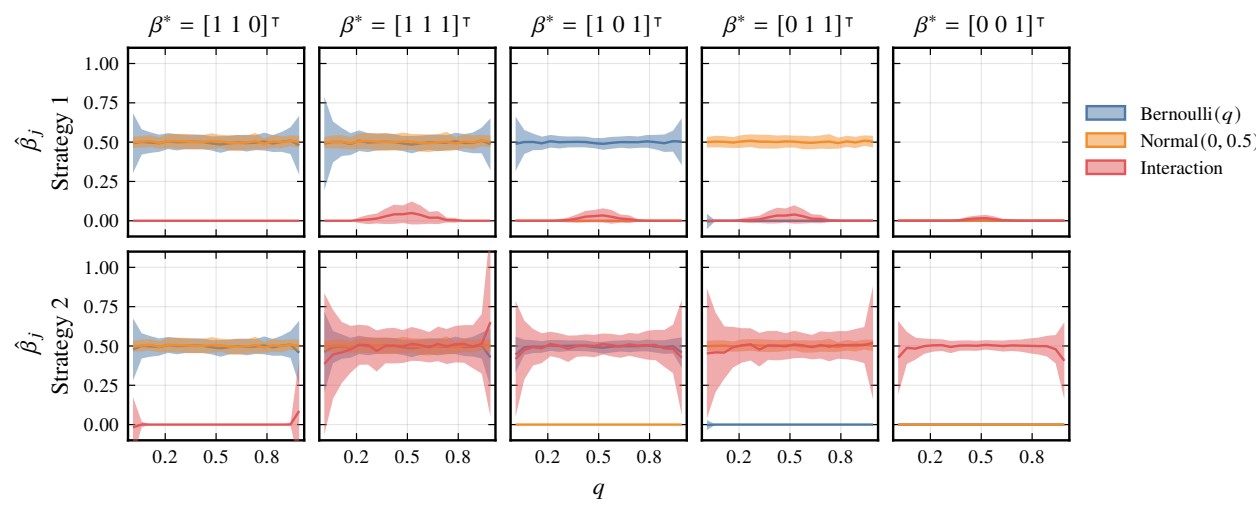

*Figure 19.* Lasso estimates for a three-feature problem where the third feature is an interaction term between the first two features. The first feature is binary with class balance $q$ and the second is quasi-normal with standard deviation 0.5. The signal-to-noise ratio is 1. The experiment was run for 100 iterations and we aggregate and report means across all iterations. Please see Section 4.1.4 for information about the different strategies.

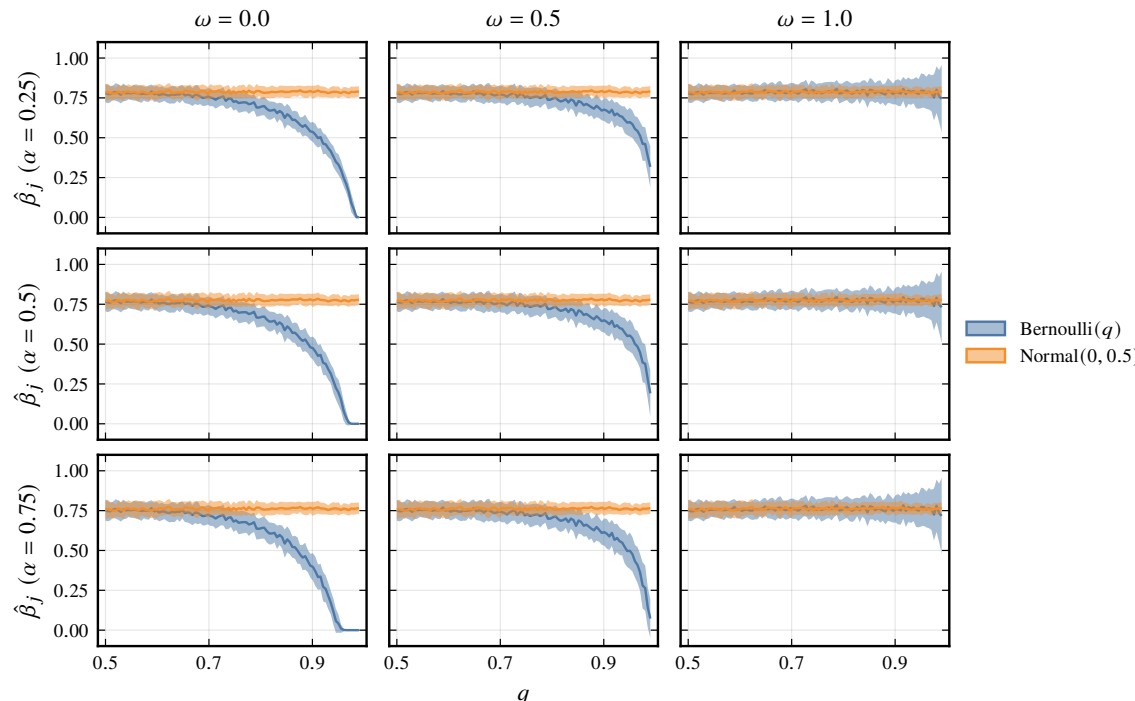

*Figure 20.* Weighted elastic net estimates for a two-dimensional problem where one feature is a binary feature with class balance $q$ (Bernoulli($q$)), and the other is a quasi-normal feature with standard deviation $1/2$ (Normal$(0, 0.5)$). Here, we have $n = 1000$ observations. The signal-to-noise ratio is 0.5. In every case, we standardize the normal feature. The binary feature, meanwhile, is centered by its mean and scaled by $(q - q^2)^\delta$. The experiment was run for 100 iterations and we aggregate and report means and standard deviations of the estimates.

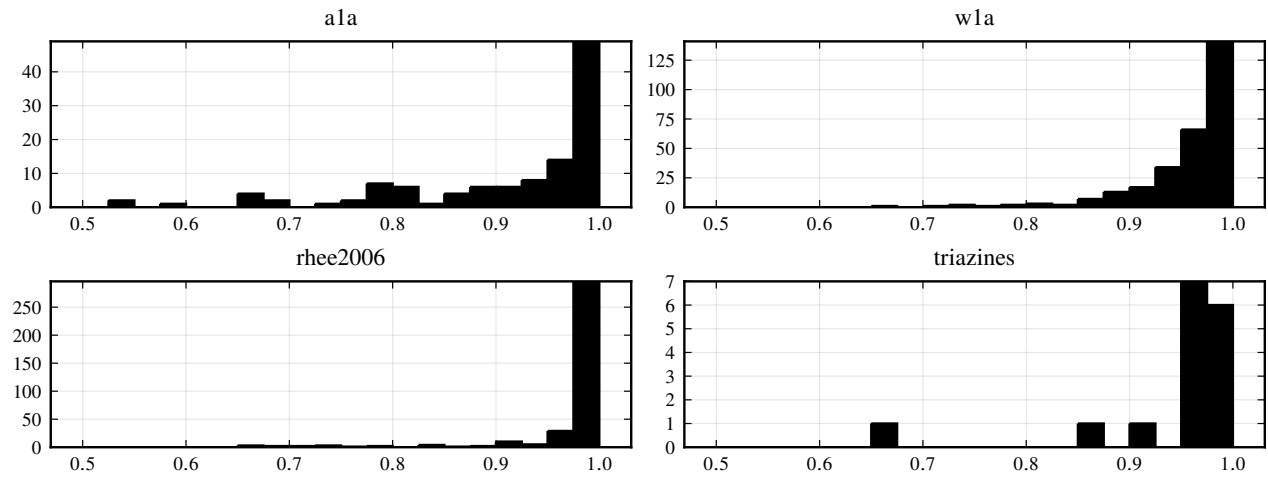

*Figure 21.* Histograms over the distribution of $q$ (class balance, that is, the proportion of ones) for the binary features in each of the data sets used in the paper.

