# OpenReview forum: "The Choice of Normalization Influences Shrinkage in Regularized Regression"
_ICML.cc/2025/Conference — Submitted to ICML 2025_

### Official Review · Reviewer_dUhp · 2025-03-08

**Overall Recommendation:** 4

**Summary:**

The paper proposed to study the nature and impact of feature normalization schemes with respect to linear models under the L1, L2, and Elastic-Net penalties. This is done only for regression, and is focused particularly on binary features. The results are primarily theoretical in nature, with a limited number of datasets evaluated.



---- Post Rbuttal Update ---

Having read the other reviews, I do not believe their concerns are well founded. There are stylistic concerns that are noisy factors, I obviously found the paper well written enough with some suggestions that the authors seem to have taken hernestly - a difference in perception on subjective factors is no reason to reject a work.

Similar concerns about ImageNet show a lack of familiarity with the breadth of ML research and what solves problems in the real world. I have used $L_1$ penalized models over decades to make more effective, faster, and cheaper solutions to real-world problems deployed across the globe. Deep learning is not the beginning or the end of AI/ML.

This paper takes a refreshing new perspective to provide theory and insights to an often overlooked matter in feature normalization. We should reward such creativity, which can have a real-world impact, and spawn new avenues for research.

**Claims And Evidence:**

Though the claims do have evidence, the abstract set my expectations considerably beyond what the article contains. I had anticipated
1. Logistic models to be considered as well
2. A larger collection of datasets
3. An empirical evaluation of "our recommendations" vs the readily available tools

**Essential References Not Discussed:**

No critical missing references, but it would be nice to give pointers to the many libraries that have these models to highlight impact (Scikit, DLIB, JSAT, Celer, and so many more).

**Experimental Designs Or Analyses:**

I have no issues with the content of what was done, but  find the most obvious experiments seem to be missing.

1. Use each of the listed normalization approaches and the current recommendation in a table for each of L1, L2, and Elastic-Net regression. Show the final total difference in predictive performance achieved using the new theoretically derived insights. If a positive improvement is shown, it also validates the acceptability of the theoretical model, assuming Gaussian errors.
2. Consider more datasets, which it is easy to binarize other datasets to match the scope of the model. In this way a statistical test via the Wilcoxon Signed Rank Test can be performed to show conclusively that the approach is an improvement.  See, Should We Really Use Post-Hoc Tests Based on Mean-Ranks? https://jmlr.org/papers/v17/benavoli16a.html

**Methods And Evaluation Criteria:**

The theoretical approach is sound as far as I can tell. Many figures would be clearer, especially the ones with simulated results, by instead centering the plots to the deviation from the estimated effect $\hat{\beta}_j$ from the known true effect $\beta_j$, as it is otherwise difficult to tell at a glance what the results mean and how to interpret them. Indeed while well written, every result is presented in a way that presumes the reader is intimately familiar with the larger statistical literature. I think the paper is suffering a bit from "I wrote it and I know it", and could use a friendly pass from a colleague unaware of the work previously.

**Other Comments Or Suggestions:**

The article would be improved with some more guidance to the reader of "where we are going and why", each section currently "jumps in" and it is not clear what point is necessarily being made until the end. That isn't to say that the paper isn't good grammar, but it was my excitement that carrier me through - it would have been more pleasant if I had a map!

Considering or discussion robust standardization methods would also be appreciated, but a weakness I could accept if other items were addressed.

**Other Strengths And Weaknesses:**

I was very excited about this paper initially. I've solved many real-world problems with L1 penalized models, followed by Xgboost, and that gets you 90% of the world. Briding the empirical gap to show that the results are practical in a straight up, "what is the Accuracy /AUC", as mentioned in the above sections, would massively elevate the quality and impact of the paper.


The paper's readability would be greatly improved by including an Algorithm block for each of L1, L2, and Elastic-Net of the author's proposed approach to normalization, each is currently buried in the text and hard to separate in the content.

**Questions For Authors:**

See above content, if you could provide:

1. Larger experimental evaluations
2. Algorithm blocks for each case

I will raise my score to accept and strongly champion the work, though I would still encourage improvements in writing.

**Relation To Broader Scientific Literature:**

Linear models make the real world go round, the impact could be massive. Most literature in this space consider the algorithm independent of the data normalization (or just pick something). Even in other areas like Differential Privacy, it is standard to assume a specific L1 normalization of the data for theoretical reasons, and this work may evidence som understandings of an implicit difference between DP and standard regressions beyond the effect of randomized noise.

**Theoretical Claims:**

I have not manually checked the proofs.

---

> ### Author Rebuttal · Authors · 2025-03-31
>
> Thank you for your extensive review of our paper. We appreciate the time and
> effort you have put into providing feedback and hope that our responses
> will address your concerns. We will start by addressing your comments
> regarding the experimental design and request for additional experiments.
>
> ## Extended Experiments and Comparisons
>
> If we understand correctly, then you ask for a new experiment
> where we compare the different normalization methods on an extended suite of
> real data sets. We have run such an experiment, which now also
> includes results with logistic loss, and present the results
> here: <https://imgur.com/a/hRAw7lf>. They show that our method (tuning over
> the normalization parameter $\delta$) performs best among all the methods
> evaluated here. We would like to make the
> following caveats regarding this experiment, however:
>
> - Our paper is not primarily focused on predictive performance, but
>   rather on estimation accuracy and feature selection. As we have shown,
>   normalization has a significant impact on prediction as well, but this is not
>   the main focus of our work.
> - As we state in our paper, the elastic net requires special consideration and
>   would necessitate an altogether different approach. While not impossible, we
>   would be happy to do so but will not have time to finish this for the review.
> - We are not sure that the Wilcoxon Signed Rank Test is the best
>   choice for this kind of comparison. The symmetricity assumption will not be
>   met in our case and the test would ignore the magnitude of the differences,
>   which we believe to be important in our case.
>
> ## Binarization Experiment
>
> We have conducted an experiment on the effect of normalization when
> binarizing continuous features. To dichotomize in a meaningful
> way, we used contextual information from the data set, for example
> dichotimizing
>
> - the crime rate variable by the national average (for 1970-1971),
> - the NOx variable by the EPA standard (at the 1971 level), and
> - the presence/absence of large lot zoning.
>
> We fit the lasso path to each of the binarized data sets as
> well as OLS. Then we compared the ranks of the OLS estimates
> to the order in which the features appear in the lasso path and
> computed correlation measures. The results are shown
> here: <https://imgur.com/a/FNugwMx>. As you can see,
> using $\delta=1$ (scaling with variance) yields best correspondence
> between the OLS estimates and feature selection order. We will include
> these results in the final version of the paper.
>
> ## On Methods and Evaluation Criteria
>
> ### Plot Presentation
>
> Thank you for suggesting centering plots on deviation from estimated effects.
> We've considered this carefully but believe the current presentation better
> serves reader comprehension since:
>
> - We show regularized estimates, not true coefficients (which are 1 in the
>   experiment).
> - Centering would therefore center most effects around -0.5, making shrinkage
>   interpretation less intuitive.
> - When estimates shrink to 0, a centered plot would show -1, requiring readers
>   to know the true value is 1.
>
> We experimented with your suggestion (see:
> <https://imgur.com/a/bGJ1eNQ>) but find the original approach more clear.
>
> ### Accessibility Improvements
>
> We've made substantial revisions to make our paper more accessible to readers
> less familiar with the statistical literature, including clearer terminology,
> additional context, and improved explanations of key concepts.
>
> ## On Broader Scientific Context
>
> We appreciate your recognition that our work could significantly impact how
> practitioners approach regularized modeling. Linear models remain fundamental
> across many domains, and understanding normalization effects addresses a
> critical gap in the literature.
>
> ## On Essential References not Discussed
>
> Following your suggestion, we've added a new paragraph discussing popular
> libraries that implement these
> models, highlighting that our findings may potentially have major consequences for a large and broad group of people.
>
> ## Practical Guidelines
>
> While our paper doesn't introduce new algorithms per se, we recognize the need
> for clear recommendations. We've added a structured section presenting
> actionable guidelines for practitioners on:
>
> - How to select appropriate normalization methods for different regularization
>   techniques
> - When particular normalization approaches may be preferred or avoided
> - Considerations for data with imbalanced binary features
> - Practical impact on coefficient interpretation
>
> ## Paper Structure Improvements
>
> We've enhanced the paper's flow and readability by:
>
> - Adding clearer "signposting" at section beginnings to orient readers
> - Improving transitions between theoretical concepts and practical implications
> - Providing better context for technical derivations
> - Including summary points that connect individual findings to our broader
>   argument
>
> ## Additional Topics
>
> We've incorporated a brief discussion of robust standardization methods in the
> discussion section as suggested.

---

> > ### Comment · Reviewer_dUhp · 2025-04-02
> >
> > While there isn't a dramatic "algorithm" perse, it would still be useful to have pseudo-code in a latex algorithm environment for "if you want to code this yourself, there's are the relevant equations/steps in the correct order". Please include it, it will help with the "Practical guidelines" and I think elevate this work.

---

> > > ### Author Response · Authors · 2025-04-02
> > >
> > > Thank you for your clarification. We recognize that this could be helpful for the reader and have added an algorithm to clearly state how we normalize binary vs continuous features in the paper.

---

### Official Review · Reviewer_YpvQ · 2025-03-13

**Overall Recommendation:** 2

**Summary:**

This paper investigates how different normalization strategies affect the shrinkage in regularized regression models such as Lasso, Ridge, and Elastic Net. The authors analyze the impact of normalization on binary and continuous features, noting that class balance directly affects regression coefficients and that different normalization methods may introduce a trade-off between bias and variance. The paper proposes the possibility of replacing feature normalization with a weighted Elastic Net approach. Experimental results are validated using both synthetic and real data.

**Claims And Evidence:**

The claims are overall supported by theoretical or experimental evidence.

**Essential References Not Discussed:**

I cannot judge whether the proposed method/analyses is  novel and significant in the lasso problem, since I am not familiar to the topic of Lasso

**Experimental Designs Or Analyses:**

The experiments provided metrics for various parameters; it would be helpful to include accuracy comparison results.

This paper should conduct more experiments on large-scale classification tasks.

**Methods And Evaluation Criteria:**

(1)The evaluation is based on synthetic and small-scale data, which is far from the real scenario. I understand that this paper is only for simple (linear) model. However, the sparsity property is also somewhat important for deep neural network. I think this paper should tailor the method and analyses for deep neural networks. Otherwise, I cannot find its value in practice. (I cannot judge whether the proposed method/analyses is enough novel and significant in the lasso problem, since I am not familiar to the topic of Lasso).


(2)The paper analyzes traditional regression models, but current predominant models also include CNNs, Transformers, etc., which use normalization techniques such as Batch Normalization and Layer Normalization. It would be valuable to provide some discussion and experimental results on the applicability of the conclusions to these models and normalization methods.

**Other Comments Or Suggestions:**

Formula Derivation Confirmation: In Equation 12, how is $ x_j^T ε $  derived? Should it be $ \tilde{x}_j^T ε$  instead? A detailed derivation process would be appreciated.

**Other Strengths And Weaknesses:**

The paper provides mathematical derivations analyzing the effects of normalization on coefficient estimation, bias, and variance in Lasso, Ridge, and Elastic Net models.

**Questions For Authors:**

NA.

**Relation To Broader Scientific Literature:**

I cannot judge whether the proposed method/analyses is  novel and significant in the lasso problem, since I am not familiar to the topic of Lasso

**Theoretical Claims:**

I do not find remarkable errors in theoretical claims.   My main concern is the assumption of feature orthogonality.  The theoretical analysis assumes feature orthogonality, which is often not the case in real data. It would be beneficial to include analysis and experimental results for these scenarios.

---

> ### Author Rebuttal · Authors · 2025-03-31
>
> Thank you for your review of our paper. We appreciate the time and effort you
> have put into providing feedback and hope that our responses to your comments
> will address your concerns.
>
> > ## Claims And Evidence
> >
> > (1)The evaluation is based on synthetic and small-scale data, which is far from
> > the real scenario. I understand that this paper is only for simple (linear)
> > model. However, the sparsity property is also somewhat important for deep
> > neural network. I think this paper should tailor the method and analyses for
> > deep neural networks. Otherwise, I cannot find its value in practice. (I cannot
> > judge whether the proposed method/analyses is enough novel and significant in
> > the lasso problem, since I am not familiar to the topic of Lasso).
>
> We respectfully disagree that the results need to be tailored to deep neural
> networks. While we appreciate the suggestion to explore connections to deep learning,
> our focus on regularized linear models addresses an important and foundational issue
> in statistical learning that has been overlooked in the literature. As we demonstrate
> throughout the paper, feature normalization has significant impacts on parameter
> estimates and model selection even in these simpler settings, which are still widely
> used in many practical applications including interpretable AI, biostatistics, and
> economics. Furthermore, understanding these fundamental behaviors in linear models provides
> essential groundwork for extending similar analyses to more complex models in the future.
>
> > (2)The paper analyzes traditional regression models, but current predominant
> > models also include CNNs, Transformers, etc., which use normalization
> > techniques such as Batch Normalization and Layer Normalization. It would be
> > valuable to provide some discussion and experimental results on the
> > applicability of the conclusions to these models and normalization methods.
>
> We agree that it would be interesting to investigate batch and layer
> normalization for neural networks and the connection to regularization and are
> happy to add a remark regarding this in the paper's discussion. Tackling this
> issue directly in the paper, however, would require a completely different
> analysis and empirical evaluation, which makes us think that this is better
> suited for future work.
>
> > ## Theoretical Claims
> >
> > I do not find remarkable errors in theoretical claims. My main concern is the
> > assumption of feature orthogonality.
>
> Please see the response to reviewer nR8u regarding the assumption of orthogonality.
>
> > ## Experimental Designs Or Analyses
> >
> > The experiments provided metrics for various parameters; it would be helpful to
> > include accuracy comparison results.
> >
> > This paper should conduct more experiments on large-scale classification tasks.
>
> Thank you for this suggestion. By "accuracy," we assume you mean predictive performance.
> We have included results on additional data sets and also included results on
> classification, which we have added to the supplementary material. See the
> response to reviewer dUhp for details and plots of the new results. We have also
> updated the dimensions of the experiments in section D.2.1 to include more
> features as well as observations, but this makes no difference for our results.
>
> > ## Other Comments or Suggestions
> >
> > Formula Derivation Confirmation: In Equation 12, how is $x_j^T \varepsilon$
> > derived? Should it be $x_j^T \varepsilon$ instead? A detailed derivation
> > process would be appreciated.
>
> Thank you for pointing this out. There is indeed a mistake in the equation, although
> it should not in fact be $\tilde{\boldsymbol{x}}_j$. Instead, the term
> $c_j \mathbf{1}^\intercal \boldsymbol{\varepsilon}$ should be present in the numerator
> as well, which we have now corrected. This correction doesn't affect the result since
> the expectation of this term is 0.

---

> > ### Comment · Reviewer_YpvQ · 2025-04-07
> >
> > Thanks for the response to my comments.  I still have concerns on the  assumption of feature orthogonality. Does the authors can conduct experiments on real datasets with the assumption of feature orthogonality?
> >
> > Besides,  I still cannot find the value  of this paper in practice.  I know  "**feature normalization** has significant impacts on parameter estimates and model selection even in these simpler settings" and widely used in deep learning (normalizing the activations, e.g., batch normalization, layer normalization). But my concern is "the evaluation is based on synthetic and small-scale data, which is far from the real scenario". In another way, does the experiments can be conducted on ImageNet-1000 classification? or object detection tasks on Coco? or maybe, providing an experiments on the "Interpretable AI, biostatistics, and economics" as you mentioned in the rebuttal?

---

> > > ### Author Response · Authors · 2025-04-08
> > >
> > > > I still have concerns on the assumption of feature orthogonality. Does the authors can conduct experiments on real datasets with the assumption of feature orthogonality?
> > >
> > > Please see the response to reviewer nR8u regarding the assumption of feature normalization, where we discuss the assumption in detail as well as provide a new experiment that shows that this assumption has, at least empirically, no effect on our results. We are not certain if you meant to write "with" or "without" regarding the assumption of feature orthogonality. If you meant "without", and are asking us to present results on data sets without the assumption of orthogonality, then please note that we already have several experiments on real, non-orthogonal, data and have shown that the effect of normalization is strong even in these cases as well. Also see the response to reviewer dUhp, where we present new results on additional data sets and show that hyper-parameterizing over normalization has real effects for predictive performance as well as feature selection and estimation. If you ask for data sets with the assumption of orthogonality, however, then note that our theoretical results cover this scenario exactly.
> > >
> > > > But my concern is "the evaluation is based on synthetic and small-scale data, which is far from the real scenario".
> > >
> > > We would respectfully like to point out that this is not correct. Our evaluation is not only based on synthetic data. Neither is it far from the real scenario. Our real data examples span a wide range of different tasks and types of data structures, including mixed binary and continuous features, only binary features, and sparse and dense data. They also span several different domains, such as socio-economics, bioinformatics, web-browsing, electricity consumption, credit ratings, and more. We have also, during the review, expanded our results to both classification and regression.
> > >
> > > > In another way, does the experiments can be conducted on ImageNet-1000 classification? or object detection tasks on Coco? or maybe, providing an experiments on the "Interpretable AI, biostatistics, and economics" as you mentioned in the rebuttal?
> > >
> > > While ImageNet and Coco would be a useful data sets to examine in the context of image recognition, they would not be suitable for the models we are considering here: the lasso, ridge, and elastic net. As we pointed out earlier, the results in our paper are robust to increasing dimensions in our experiments. That being said, however, we do recognize that it would be useful to include additional experiments on real data sets. Therefore, we have now run experiments on additional data sets here of larger sizes, and show the results in the table below, which show the combination of $\lambda$ and $\delta$ with the lowest test set error. As you can see, the best value for $\delta$ differ depending on the data set. The setup of the experiment is the same as in Section 4.1.2 in our paper, but we have used a grid of 11 $\delta$ values instead of 100.
> > >
> > > | Dataset | $n$ | $p$ | $\delta$ | $\lambda$ | NMSE |
> > > |---------|-------------|----------|-------|--------|-------|
> > > | YearPredictionMSD | 463,715 | 90 | 0.0 | 0.000251189 | 0.761951 |
> > > | covtype.binary | 581,012 | 54 | 0.3 | 0.0001 | 0.701364 |
> > > | rcv1_train.binary | 20,242 | 47,236 | 0.0 | 0.01 | 0.986731 |
> > > | real-sim | 72,309 | 20,958 | 0.9 | 0.158489 | 0.999964 |
> > >
> > > We would be happy expand this list with additional, and even larger, data sets for the final version but will unfortunately not have time to do this before the review period ends.

---

### Official Review · Reviewer_nR8u · 2025-03-16

**Overall Recommendation:** 2

**Summary:**

This paper studies the effects of input normalization for LASSO, Ridge, and ElasticNet regression, focusing on normal and binary features. See below for more detailed discussions on the settings and contributions of this paper.

**Claims And Evidence:**

yes.

**Essential References Not Discussed:**

N/A.

**Experimental Designs Or Analyses:**

Yes. In Section 4.1.1, the authors explain how class balance affects coefficients in LASSO with synthetic data, showing how the estimation varies with different normalization (no scaling vs. variance scaling). In Section 4.1.2, the authors investigate how normalization affects the predictive performance with three real datasets. In Section 4.1.3, the authors demonstrate how to normalize when we have both binary and continuous features. In Section 4.2, the authors present an alternative way to normalize: weighting the regularization terms instead of normalizing data.

**Methods And Evaluation Criteria:**

yes.

**Other Comments Or Suggestions:**

### General comment
Overall, I find the problem being considered interesting. However, I find the presentation of this paper a bit underwhelming, and I feel uncomfortable reading this paper. I believe that the presentation can be greatly improved. However, in this form, I believe this draft is not ready to be published, and I cannot recommend acceptance for this paper. But don't worry, I will consider updating my score after the authors make changes to improve the presentation, clarity, and flow of this paper. Here are some suggestions:

### Minor changes: Potential typos/clarification suggestions
1. In Equation 2, $\\tilde{\\boldsymbol{X}}$ is used before defined in Definition 2.1. Consider adding a small clarification right after Equation 2, even if it is repetitive.
2. In the first paragraph of Section 3, there is no definition for the noise $\\varepsilon_i$ (it is defined later). Consider adding this clarification right after using it.
3. In the first paragraph of Section 3, maybe explicitly mentioning that $\\boldsymbol{x}_j$ is the $j$-th column of $\\boldsymbol{X}$ could improve clarity.
4. The same goes with $\\tilde{\\boldsymbol{x}}_j$.
5. Please discuss the (strong) assumption on the orthogonality of the normalized design matrix $\\tilde{\boldsymbol{X}}$: (1) how it rarely holds in practice, (2) how it is necessary in this framework, (3) if any prior works also have this assumption (to see that it is not too weird one).
6. In the first paragraph of Section 3.1., what is $\\overline{\\boldsymbol{x}}? I know that it is defined in Table 1, but please refer to it once again to help the readers keep track of the notations. Besides, the wording of this paragraph is not good; consider rewriting this paragraph.
7. What is $\\Phi(\\cdot)$ in Equation 8? I know this one is the CDF of normal distribution, but please define it clearly here.
8. In section 4.1.2, the author should at least describe what those datasets (a1a, rhee2006, and w1a) are about. I know that the statistics of that dataset is mentioned in Appendix E, Table 2, but at least the author should refer to that Table in the main body.
9. In Theorems 3.1, 3.2, and 3.3, since $q_j \\in [0, 1]$, why do the authors write the limit $q_j \\rightarrow 1^+$? Do you mean $1^-$ instead?

### Major changes: paragraph/section suggestions
1. It is better if the authors write the whole paragraph/sections of notations used in this paper. The way the authors use notations here is not good: it makes the readers have a hard time finding relevant notions.
2. Consider adding the contributions paragraph at the end of Section 1. List the main contributions clearly and where they are located in the main body. I know that the authors discuss those points in Section 5, but I think it is better if authors have some punch lines right from the start so that the readers can keep track of the paper more easily. Consider reorganizing Section 5 accordingly, making it more condensed, and putting the punch lines right at the start.
3. Section 3 is very monotone; please reorganize this section. For example, in Section 3.1, the authors could make each part of Section 3.1 more distinguishable, for example, by writing:

    (1) "\paragraph{Class blance}", which contains the rigorous definition and its intuition.

    (2) "\paragraph{the effects on the estimators scale ...}" discusses how the class balance directly affects the estimator.  Please make three paragraphs after Equation 7 stand out more and elaborate on how the finding is interesting and counterintuitive.

    (3) and so on.

The above are just some suggestions that I have. I recommend that the authors do multiple passes to improve the presentation of this draft.

**Other Strengths And Weaknesses:**

## Strength
I like the problem considered here, and I agree that this problem is somewhat under-investigated and worth pursuing.

## Weaknesses

1. This paper is not well-polished. The flow is very monotone, making the readers have a hard time reading this paper. See below for suggestions.

2. The assumption on the orthogonality of the normalized design matrix $\\tilde{\\boldsymbol{X}}$ is super strong and rarely holds in practice. This is even more unrealistic for two cases that this paper considers here: binary features and mixed features. As a result, it is very hard to tell if the findings in this paper have an interesting implication in the general cases and in practice.

3. As a consequence, I expect that it is the technical challenge on the theoretical side that makes the paper more interesting. However, this is not the case here.

**Questions For Authors:**

1. Is the assumption of the orthogonality of the normalized design matrix too strong and unnatural, especially for the case of binary features and mixed features? Please give me a couple of examples where this assumption holds and a couple of prior works that deal with this assumption. My intuition is that this assumption is extremely weird in the cases that this paper considers.

**Relation To Broader Scientific Literature:**

N/A.

**Theoretical Claims:**

Yes. Under strong assumptions, the authors make the following claims:

1. In Theorem 3.1, when the class balance becomes very extreme ($q_j \\rightarrow 1^-$ (there is a problem in the draft, where the authors write $q_j \\rightarrow 1^+$)), the coefficients of the estimators depend heavily on the normalization parameters.

2. In Theorem 3.2, when the class balance becomes very extreme, the variance of the coefficients of estimators also strongly depends on the normalization parameter.

3. In Theorem 3.3, they have analogous results to Theorems 3.1 and 3.2, but for the weights of regularizations.

---

> ### Author Rebuttal · Authors · 2025-03-31
>
> Thank you for your detailed review of our paper. We appreciate the time and
> effort you have put into providing feedback. We will start by addressing
> your comments regarding the assumption of orthogonality.
>
> ## Assumption of Orthogonality
>
> We agree that the assumption is strong and unrealistic. Nevertheless, it is not
> uncommon in literature, particularly not in the context of research on
> regularized methods, where the theoretical analysis is often difficult. Please
> see the following references for examples where the authors make similar (and
> sometimes even stronger) assumptions:
>
> - Efron, B., Hastie, T., Johnstone, I. M., & Tibshirani, R. (2004). Least angle
>   regression. Annals of Statistics, 32(2), 407–499.
> - Fan, J., & Li, R. (2001). Variable selection via nonconcave penalized likelihood and
>   its oracle properties. Journal of the American Statistical Association, 96(456),
>   1348-1360.
> - Bogdan, M., Berg, E. van den, Sabatti, C., Su, W., & Candès, E. J. (2015).
>   SLOPE – adaptive variable selection via convex optimization. The Annals of
>   Applied Statistics, 9(3), 1103–1140.
> - Yuan, M., & Lin, Y. (2005). Model selection and estimation in regression with
>   grouped variables. The Journal of the Royal Statistical Society, Series B
>   (Statistical Methodology), 68(1), 49–67.
> - Bu, Z., Klusowski, J., Rush, C., & Su, W. (2019). Algorithmic analysis and
>   statistical estimation of SLOPE via approximate message passing. In H. Wallach,
>   H. Larochelle, A. Beygelzimer, F. dAlché-Buc, E. Fox, & R. Garnett (Eds.),
>   Advances in neural information processing systems 32 (pp. 9361–9371). Curran
>   Associates, Inc.
>
> We would also like to argue that it is not unreasonable to begin with
> a simple setting, especially given that this is the first work to investigate
> this issue and that, even in this simple setting, the effect of normalization
> is strong.
>
> More importantly, however, we believe that it is reasonable
> to assume that the assumption of orthogonality is not in fact
> restrictive for our results. See, for instance, the experiment presented in
> figures 4 and 15, where we have introduced correlation between the features and
> where the effect of class-imbalance is _stronger_ when the features are
> correlated. We have also for this review conducted a new
> experiment to investigate this assumption further, which we
> present in the figure in the link below. Here, we have studied the case
> of two binary features with varying levels of correlation
> and let the class balance of the second feature
> tend to 1. The results show that the effect of class imbalance
> is unaffected by correlation.
>
> <https://i.imgur.com/RzijpY6>
>
> We realize, however, that the paper would benefit from a more thorough
> discussion of this assumption, and we will include this experiment along with a
> detailed discussion of the assumption in the final version of the paper. We will
> also add a new theorem that shows that correlation between two features
> shrinks to zero in the limit if one of the features is binary and its class balance tends
> to 1 (or 0).
>
> ## Impact
>
> > I like the problem considered here, and I agree that this problem is somewhat
> > under-investigated and worth pursuing.
>
> Thank you for your appreciation regarding the impact and value of the problem
> considered in the paper. We agree that it is under-investigated and worth
> pursuing.
>
> ## Notational Remarks and Other Minor Suggestions
>
> Thank you for your detailed comments on the notation and suggestions on
> clarifications. We have made the changes you suggested and believe that the
> paper is now clearer and more accessible to a wider audience.
>
> ## Writing and Presentation
>
> We are sorry to hear that you found the paper hard to read. We have
> now restructured the paper along your recommendations, including adding
> a section on notation and a summary of main contributions,
> and believe that the paper now flows better and is easier to read.
>
> ## Details on Data Sets
>
> > 8. In section 4.1.2, the author should at least describe what those datasets
> >    (a1a, rhee2006, and w1a) are about.
>
> We have added a reference to the appendix and detailed descriptions of these
> data sets in the section in the appendix.

---

### Decision · Program_Chairs · 2025-05-01

**Decision:**

Reject

**Comment:**

There were two relatively critical reviews and one high positive. To start with the latter, the main arguments in favor of the paper seem to be (i)  the importance, relevance and novelty of the research question posed -- namely the influence of feature normalization on shrinkage  in penalized linear regression models, and (ii) the theoretical soundness of the analysis.
The two more critical reviewers raised concerns about unrealistic assumptions (like orthogonal features), lacking comparison experiments in large-scale real-world situations and a somewhat narrow focus on linear models with Gaussian likelihoods and (mostly) binary inputs.

After going through the paper, all the reviews, rebuttals and discussions again, I am not fully convinced by the arguments in the one very positive review. Neither do I agree that the research question about the influence of data normalization is particularly novel, nor do I think that the theoretical analysis provided is very helpful. I would rather argue that the problems of normalizing real-valued inputs and encoding categorical inputs are among the most classical problems studied in the literature on linear models. Further, I honestly have never seen a realistic high-dimensional dataset, where the assumption of (even approximate) feature orthogonality could have been justified in practice: typically the empirical covariance matrix is far from being diagonal or at least diagonally dominant.   I am also not convinced by the authors' comment that similar assumptions have been used in other papers, because most other relevant papers take a more general perspective by using the orthogonal case only as a starting point in their analysis, but also considering correlations. In summary, I fully agree with the critical comments about unrealistic assumptions, and I also share most of the critical comments on limited and unconvincing experimental validation. Therefore, I tend to recommend rejection of this paper.